# LAMP-coupled CRISPR-Cas12a assays: A promising new tool for molecular diagnosis of leishmaniasis

Eva Dueñas[1¤a☯], Ingrid Tirado[1,2☯], Percy Huaihua[3], Ariana Parra del Riego[1], Luis Cabrera-Sosa[1,4¤b], Jose A. Nakamoto[1¤c], María Cruz[5], Carlos M. Restrepo[6], Jorge Arévalo[3,4], Vanessa Adaui [1,4]*

1 Laboratory of Biomolecules, Faculty of Health Sciences, Universidad Peruana de Ciencias Aplicadas (UPC), Lima, Peru, 2 Facultad de Ciencias Biológicas, Universidad Nacional Mayor de San Marcos, Lima, Peru, 3 Laboratorio de Patho-antígenos, Laboratorios de Investigación y Desarrollo, Facultad de Ciencias e Ingeniería, Universidad Peruana Cayetano Heredia, Lima, Peru, 4 Instituto de Medicina Tropical Alexander von Humboldt, Universidad Peruana Cayetano Heredia, Lima, Peru, 5 Facultad de Ciencias de la Salud, Universidad Andina del Cusco, Cusco, Peru, 6 Centro de Biología Molecular y Celular de Enfermedades, Instituto de Investigaciones Científicas y Servicios de Alta Tecnología (INDICASAT AIP), Panama City, Panama

☯ These authors contributed equally to this work.
¤a Current Address: Life Science Research Centre, Faculty of Science, University of Ostrava, Ostrava, Czech Republic
¤b Current Address: Department of Biomedical Sciences, Institute of Tropical Medicine Antwerp, Antwerp, Belgium
¤c Current Address: Laboratory of Protein Evolution, Department of Experimental Medical Science, Lund University, Lund, Sweden
* vanessa.adaui@upc.pe

## Abstract

### Background

Tegumentary leishmaniasis is a parasitic disease endemic in the Americas. Its clinical management and control rely on early and accurate diagnosis and adequate treatment. PCR-based molecular diagnostics offer high sensitivity and specificity over microscopy or culture but are less accessible in low-resource settings. New molecular tools for detecting *Leishmania* infections are needed in rural endemic regions. A promising tool harnessing CRISPR-Cas technology enables highly specific and sensitive detection of nucleic acid targets, offering an exciting potential for portable molecular diagnostics. Previously, we developed CRISPR-Cas12a-based assays coupled to PCR preamplification for *Leishmania* detection. Here, we adapted our assays, which target the multicopy 18S rDNA and kinetoplast DNA (kDNA) minicircles, by replacing PCR with loop-mediated isothermal amplification (LAMP).

### Methodology/Principal Findings

LAMP-coupled CRISPR assays were optimized for fluorescence-based and lateral flow readouts. The assays could detect as low as 0.2 genome equivalents per

**Data availability statement:** All relevant data are within the manuscript and its Supporting Information files.

**Funding:** This work was supported by ProCiencia, the Peruvian National Council for Science, Technology, and Technological Innovation (CONCYTEC) - The World Bank (contracts 036-2019-FONDECYT-BM-INC.INV to VA; 095-2018-FONDECYT-BM-IADT-AV to JA) and the Universidad Peruana de Ciencias Aplicadas (internal fund C-005-2024 to VA). IT was supported by a postgraduate scholarship of ProCiencia (contract PE501084865-2023). The funders had no role in study design, data collection and analysis, decision to publish, or preparation of the manuscript.

**Competing interests:** The authors have declared that no competing interests exist.

reaction using *L. braziliensis* M2904 strain genomic DNA. The kDNA assay reliably detected all tested species of the *L.* (*Viannia*) subgenus, while the 18S assay showed pan-*Leishmania* detection capability. There was no cross-reactivity with other protozoan (*Trypanosoma cruzi* and *Plasmodium falciparum*) and bacterial (*Mycobacterium tuberculosis*) pathogen DNA, nor with human DNA. When applied to 90 clinical samples (skin lesions) from the Cusco region of Peru and compared to kDNA real-time PCR, LAMP-CRISPR assays with a fluorescence readout achieved a sensitivity of 90.9% (95% CI: 80.1-97.0%) for kDNA and 72.7% (95% CI: 59.0-83.9%) for 18S rDNA, both with 100% (95% CI: 90–100%) specificity. Overall, lateral flow strip results agreed with fluorescence-based detection in 18 tested samples, with one discrepancy observed in the 18S assay associated with low parasite load.

## Conclusions/Significance

These new proof-of-concept LAMP-CRISPR assays, combining high sensitivity, multiple *Leishmania* species detection capability, and a portable lateral flow readout, hold promise as next-generation molecular tools to improve leishmaniasis diagnosis and surveillance, supporting One Health strategies for disease control.

### Author summary

Tegumentary leishmaniasis affects poverty-related populations in the Americas and encompasses skin and mucosal lesions that can cause disfigurement and social stigma. The disease is caused by several species of the protozoan parasite *Leishmania*. PCR-based molecular diagnostics are currently the most sensitive and specific diagnostic tools. Yet, these require specialized infrastructure and trained personnel that are not readily available in low-resource settings. New tools are required to meet the diagnostic needs in rural endemic areas. A promising tool leveraging CRISPR-Cas technology enables cost-effective, *in vitro* nucleic acid detection, paving the way for diagnostic solutions that could be made available to patients at, or near, the point of care. Here, we harnessed the CRISPR-Cas12a system combined with loop-mediated isothermal amplification (LAMP) to develop assays capable of detecting multiple *Leishmania* species of medical importance. Our assays employ multicopy targets widely used in molecular diagnostics: the 18S rDNA for pan-*Leishmania* detection and a kDNA minicircle region conserved among *L.* (*Viannia*) species. Results can be read with either fluorescence detection or lateral flow strips. Both assays showed satisfying performance in both analytical validation and initial clinical sample testing under laboratory conditions. These new tools show promise to improve diagnosis and surveillance of leishmaniasis.

## Introduction

Leishmaniasis is one of the most serious neglected infectious diseases, with 0.7 to 1 million new cases each year and between 20,000 and 30,000 deaths [1,2]. As a vector-borne disease, leishmaniasis poses a substantial threat to millions of individuals in endemic regions across Asia, South and Central America, Africa, and the Mediterranean basin [3]. *Leishmania*sis results from an infection with protozoan parasites of the genus *Leishmania* that are transmitted to humans (and other mammals) through the bite of infected sandflies and manifests in two main clinical forms: cutaneous (or tegumentary) and visceral leishmaniasis, each presenting unique challenges in terms of diagnosis, treatment, and control [4,5]. In the Americas, tegumentary leishmaniasis is distributed in 18 endemic countries and comprises a spectrum of clinical conditions ranging from localized skin ulcers to severe disfiguring mucosal lesions [5]. This clinical pleomorphism is largely dependent on the immune responses of the human host as well as on factors intrinsic to the different infecting *Leishmania* species [6]. Diverse *Leishmania* species from the *Viannia* and *Leishmania* subgenera circulate in Central and South American countries, with the former being responsible for the majority of leishmaniasis cases [4]. While efforts toward the development of effective prophylactic and therapeutic vaccines against human leishmaniasis are ongoing, early and accurate diagnosis paired with timely chemotherapy remain vital in combating these diseases [7].

Current diagnostic tools for leishmaniasis include microscopy, culture, immunological and molecular methods [8]. Among the nucleic acid amplification techniques (NAATs), the polymerase chain reaction (PCR) and its variants provide the most sensitive and specific tools for the detection and identification of *Leishmania* species [9,10]. While accessible in research and reference laboratories, PCR-based methods may not be readily available in rural endemic areas since they require infrastructure, resources, and technical expertise [11]. This has led to (near) point-of-care (PoC) diagnostic tests becoming a noteworthy concept, as they would allow easier, faster and less expensive alternatives that need only minimal laboratory setup [12]. Advancements towards simplifying molecular diagnostic tools gave rise to isothermal NAATs, such as loop-mediated isothermal amplification (LAMP) and recombinase polymerase amplification (RPA), which hold potential to decentralize molecular diagnostic testing [13]. Among these, LAMP has gained prominence for its simplicity and efficacy in detecting nucleic acid sequences with high specificity. LAMP operates at a constant temperature (60–65°C), so that it can be performed with a low-cost dry bath incubator [14,15]. The LAMP reaction uses 4–6 primers that target specific DNA sequences, along with the Bst DNA polymerase, an enzyme that displays a strong strand displacement activity [14]. LAMP amplifies the target DNA producing a concatenated DNA product with stem-loop structures, yielding up to $10^9$ copies in less than an hour [14]. The results can be easily visualized, e.g., on the basis of the turbidity caused by reaction by-products [16], or with the use of sequence-specific probes [17,18]. Due to its user-friendly nature and rapidity, LAMP holds great promise as a tool for the diagnosis of infectious diseases in resource-limited settings [19]. LAMP assays have been developed for genus-, complex-, and species-specific detection of *Leishmania*, demonstrating high sensitivity and specificity with good diagnostic performance in both human cutaneous leishmaniasis (CL) and visceral leishmaniasis (VL) (reviewed in [20,21]). The Loopamp *Leishmania* Detection Kit (Eiken Chemical, Japan) has also been evaluated for CL and VL diagnosis, showing diagnostic performance comparable to PCR-based methods [22].

Clustered regularly interspaced short palindromic repeats (CRISPR)/CRISPR-associated proteins (Cas) technology has recently been harnessed as a tool for molecular detection [23], based on the discovery that once Cas12a and Cas13a enzymes bound to CRISPR RNA (crRNA) recognize specific nucleic acid targets, they become activated, thereby catalyzing cleavage of both the targets (*cis*-cleavage) as well as non-target nucleic acids (*trans*-cleavage) [24,25]. This collateral *trans*-cleavage activity can be applied for nucleic acid detection by coupling it with a reporter molecule. In fluorescence-based assays, the reporter molecule consists of a DNA or RNA oligomer carrying a fluorophore and a quencher. Upon target recognition, reporter cleavage by the activated Cas enzyme generates a detectable fluorescence signal, indicating the presence of the target nucleic acid sequence in a sample. Alternatively, visualization of the Cas detection reaction can be achieved by a lateral flow strip readout using a reporter molecule labeled with FAM or digoxin and biotin [26,27]. CRISPR-based assay platforms reported thus far enable a highly sensitive detection of

nucleic acids in the attomolar range when combined with a target preamplification step using either isothermal NAATs (e.g., DETECTR [24] and SHERLOCK [25]) or PCR-based NAATs (e.g., HOLMES [28]). Notwithstanding that this two-step CRISPR-based assay format increases assay complexity and poses the risk of cross-contamination during sample transfer to the CRISPR reaction, it is a convenient approach at the initial development stage to demonstrate the potential for further assay development [29]. More recent efforts have focused on amplification-free detection of nucleic acid targets with CRISPR, and the proof-of-concept was demonstrated for direct detection of pathogen RNA and DNA with quantitative ability [30–33].

The combination of CRISPR-Cas and LAMP technologies has advanced the progress of nucleic acid-based PoC diagnostic test development. Coupling LAMP reactions to downstream CRISPR-based detection improves specificity [19,23], since LAMP is prone to non-specific amplification [34]. Integrated LAMP-CRISPR assays have been developed for the detection of several pathogens, including viruses (e.g., SARS-CoV-2 [26,35–37], human papillomavirus [38]), bacteria (e.g., *Mycobacterium tuberculosis* [39], *Klebsiella pneumoniae* [40]), and fungi (e.g., *Verticillium dahliae* [41]). CRISPR-based assays have also been developed for the detection of parasites of medical and veterinary importance, such as the protozoa *Plasmodium* spp. [29,42], *Toxoplasma gondii* [43], *Leishmania* spp. [44–47], *Trypanosoma brucei* [48,49], and *Trypanosoma cruzi* [50]; and the trematode *Schistosoma* species [51,52], typically in combination with RPA preamplification. Applications of LAMP-CRISPR for parasite detection are increasing, as illustrated by the detection of *Schistosoma* infection in host samples [53], the *in silico* design of LAMP primers and Cas12a crRNA sequences for drug resistance genotyping of *P. falciparum* [54], the detection of *T. gondii* in environmental samples [55], and the diagnosis of acute congenital *T. cruzi* infection in infants [56].

At present, leishmaniasis diagnosis in endemic areas is impaired mainly by the lack of infrastructure, equipment, trained human resources, and low access (if any) to effective diagnostic tools. New molecular diagnostic tools that are accurate, easy to use, accessible and affordable for use in resource-limited settings are required. In previous work, we developed CRISPR-Cas12a-based assays coupled with PCR preamplification for detecting *Leishmania* spp. in clinical samples [45]. Here, we adapted these assays, which target the *Leishmania* 18S ribosomal RNA gene (18S rDNA) and minicircle kinetoplast DNA (kDNA), by replacing PCR with LAMP in the preamplification step, to facilitate the road towards the development of a nucleic acid-based near-PoC diagnostic tool for leishmaniasis. We optimized assay conditions for a fluorescence signal readout in a plate reader and for visual readout on lateral flow strips. We then evaluated the diagnostic performance of the LAMP-CRISPR assays against a kDNA real-time PCR assay using stored clinical samples from the Cusco region, where *Leishmania* species of the *Viannia* subgenus circulate [57,58].

## Methods

### Ethics statement

This study used anonymized, coded DNA samples isolated from human skin ulcerative lesions obtained by biopsy, lancet scraping, cytological brush, swab, and filter paper. All patients provided written informed consent prior to specimen collection, which included authorization for future research use of biological samples and associated clinical data. Individuals with clinically suspected CL were recruited at the Hospital Nacional Adolfo Guevara Velasco (HNAGV) in Cusco, Peru, where *L.* (*Viannia*) species –most frequently *L. braziliensis*, *L. guyanensis*, and *L. lainsoni*– circulate [57,58]. Recruitment was conducted during 2019 and 2020 as part of a collaborative study between HNAGV and the Universidad Peruana Cayetano Heredia (UPCH) (contract 095–2018-FONDECYT-BM-IADT-AV to JA, financed by CONCYTEC and The World Bank). The study protocol and informed consent (registration number 103155) were reviewed and approved by the Institutional Research Ethics Committee of the UPCH (Comité Institucional de Ética en Investigación en Humanos; approval letter 063-05-19 dated 01/30/2019, latest renewed on 05/02/2023 with letter R-149-17-23).

## Study samples

In this pilot laboratory study, we developed and evaluated two assays for *Leishmania* spp. detection using LAMP coupled with CRISPR-Cas12a. As suggested by CLSI guidelines [59], early-phase evaluation studies of qualitative diagnostic assays typically use a convenience sample of approximately 100 clinical specimens (e.g., 50 positive and 50 negative) to obtain preliminary estimates of assay performance (sensitivity and specificity). In line with this practice, a total of 90 samples were available for the current study, comprising 55 positive and 35 negative samples for *Leishmania* infection, as determined by a kDNA qPCR assay [60; see below]. We report herein the diagnostic sensitivity and specificity of the new assays, along with their 95% confidence intervals (CIs), which indicate the precision of these estimates.

## Quantitative real-time PCR (qPCR)

An in-house qPCR assay [60] targeting a kDNA minicircle region conserved among *L.* (*Viannia*) species [61] served as the reference test. The human *ERV-3* gene was amplified in parallel using the primers reported by Yuan et al. [62]; it served as an indicator of specimen adequacy and to estimate the parasite load normalized to the number of human cell equivalents per sample [60].

## DNA extraction from clinical samples prior to this study

DNA from clinical samples was isolated with the High Pure PCR Template Preparation Kit (Roche), according to the manufacturer's protocol, suspended in elution buffer and stored at -20°C until use.

## DNA samples from reference strains of *Leishmania* and other microbial pathogens

Genomic DNA (gDNA) extracted from cultured promastigotes of reference strains of *Leishmania* spp. was retrieved from the DNA biobank of the *Leishmania* research group, Molecular Epidemiology Unit, Instituto de Medicina Tropical Alexander von Humboldt (IMTAvH)-UPCH in Lima, Peru, and from the collection hosted by the Institute for Scientific Research and High Technology Services (INDICASAT-AIP) in Panama. gDNA from cultured laboratory reference strains of *Trypanosoma cruzi*, *Plasmodium falciparum*, and *Mycobacterium tuberculosis* was provided by the Infectious Diseases Research Laboratory at UPCH and INDICASAT-AIP. Strain designations and sources are listed in Table 1.

At the IMTAvH and UPCH, DNA was isolated using the High Pure PCR Template Preparation Kit (Roche, Mannheim, Germany). At the INDICASAT-AIP, the Wizard Genomic DNA Purification Kit (Promega, Madison, WI, USA) was used. DNA was isolated according to the manufacturer's protocol, suspended in elution buffer or TE buffer and stored at -20°C until use.

## LAMP primer design

The 18S rDNA gene (TriTrypDB ID: LbrM.27.2.208540; at https://tritrypdb.org/tritrypdb/) and kDNA minicircles (GenBank accession no. KY698803.1) from *L. braziliensis* MHOM/BR/75/M2904 were selected for LAMP primer design. FASTA sequences up to 200 bp downstream and upstream to previously reported crRNA target recognition sites for each gene [45] were uploaded to PrimerExplorer V5 software (Eiken Chemical Co., Ltd). Three sets of external (F3 and B3) and internal (FIP and BIP) primers were obtained for 18S rDNA, but only one for kDNA. To select the final 18S primer set, the following features were considered: the less likelihood of primer dimer formation (highest ΔG value), the 3' end stability for F2 and B2 (ΔG ≤ -4 kcal/mol), the 5' end stability for F1c and B1c (ΔG ≤ -4 kcal/mol), and the less overlap between primers and the recognition site. After that, loop primers (LF and LB) for both target genomic regions were also designed using the PrimerExplorer V5 software. Final primer sets were ordered from Macrogen Inc. (Seoul, South Korea) and are listed in the S1 Table.

**Table 1. Reference strains used in this work.**

| Strain name | WHO code¥ | Source |
|---|---|---|
| *Leishmania (Viannia) braziliensis* | MHOM/BR/75/M2904 | IMTAvH-UPCH |
| *Leishmania (Viannia) braziliensis* | MHOM/BR/75/M2903 | INDICASAT-AIP |
| *Leishmania (Viannia) braziliensis* | MHOM/PE/91/LC2043 | IMTAvH-UPCH |
| *Leishmania (Viannia) braziliensis* | MHOM/PE/91/LC2177 | IMTAvH-UPCH |
| *Leishmania (Viannia) peruviana* | MHOM/PE/90/HB86 | IMTAvH-UPCH |
| *Leishmania (Viannia) peruviana* | MHOM/PE/90/LCA08 | IMTAvH-UPCH |
| *Leishmania (Viannia) peruviana* | MHOM/PE/2005/WR-2771 | INDICASAT-AIP |
| *Leishmania (Viannia) guyanensis* | MHOM/BR/1975/M4147 | INDICASAT-AIP |
| *Leishmania (Viannia) lainsoni* | MHOM/BR/1981/M6426 | INDICASAT-AIP |
| *Leishmania (Viannia) panamensis* | MHOM/PA/2018/BD-02 | INDICASAT-AIP |
| *Leishmania (Leishmania) amazonensis* | MHOM/BR/1973/M2269 | INDICASAT-AIP |
| *Leishmania (Leishmania) mexicana* | MHOM/BZ/1982/BEL21 | INDICASAT-AIP |
| *Leishmania (Leishmania) aristidesi* | MORY/PA/1969/GML | INDICASAT-AIP |
| *Leishmania (Leishmania) major* | MHOM/SA/1991/WR-1088 | INDICASAT-AIP |
| *Leishmania (Leishmania) infantum* | MCAN/BR/1997/P142 | INDICASAT-AIP |
| *Leishmania (Leishmania) donovani* | MHOM/IN/2006/WR-2801 | INDICASAT-AIP |
| *Trypanosoma cruzi* | Sylvio X10 | UPCH |
| *Trypanosoma cruzi* | VT-1 | UPCH |
| *Trypanosoma cruzi* | Tulahuen LacZ clone C4 | UPCH |
| *Plasmodium falciparum* | HB3 | INDICASAT-AIP |
| *Mycobacterium tuberculosis* | PA155-18C | INDICASAT-AIP |

¥The World Health Organization (WHO) designation code for *Leishmania* strains is as follows: host (M for Mammalia; HOM = *Homo sapiens*; CAN = *Canis familiaris*; ORY = *Oryzomys capito*)/country of origin (BR = Brazil; BZ = Belize; IN = India; PA = Panama; PE = Peru; SA = Saudi Arabia)/year of isolation/name of strain.

To investigate the *in silico* specificity of the designed LAMP primers, we downloaded all publicly available DNA sequences for each target from NCBI in May 2025. For *Leishmania* kDNA minicircle sequence analysis, a total of 1,060 sequences from the *L. (Viannia)* subgenus and 965 sequences from the *L. (Leishmania)* subgenus were retained and analyzed following a quality filtering process. Sequences that were excluded contained ambiguous base pairs (e.g., 'N') exceeding 10% of the total length or did not cover the primer binding sites. The eight LAMP primer binding regions and the crRNA target site were manually annotated using Jalview v2.11.4.1 [63] following multiple sequence alignment with Clustal Omega [64]. Sequence alignments are available in the S1 File. A representative subset of 59 sequences (24 *Viannia*, 35 *Leishmania*) is illustrated in S1 Fig to indicate sequence conservation or variations across the two analyzed *Leishmania* subgenera. As for the 18S rDNA, 37 sequences from *Leishmania* spp. and 272 from *Trypanosoma* spp. were aligned and analyzed using the same bioinformatics pipeline (S2 File). A subset of 31 representative sequences (16 *Leishmania* spp., 15 *Trypanosoma* spp.) was selected to denote sequence conservation or variations across the two analyzed trypanosomatid genera (S2 Fig).

### crRNA guide sequences and preparation

We used previously reported LbCas12a crRNA guide sequences targeting 18S rDNA and kDNA minicircle sequences [45]. crRNA preparation from double-stranded DNA templates (see S1 Table for nucleotide sequences) was performed as described [45].

## Loop-mediated isothermal amplification (LAMP) assay

We first optimized the LAMP assay reaction conditions using Bst 2.0 DNA polymerase (New England Biolabs Inc. (NEB), Massachusetts, USA; Cat no. M0537) according to the manufacturer's protocol. The only modification was the final total concentration of $MgSO_4$ in the reaction (6 mM $Mg^{2+}$ for 18S rDNA and 5 mM $Mg^{2+}$ for kDNA). These reaction conditions were used to evaluate the incubation time needed for optimal amplification of the target DNA, which was tested at $2 \times 10^2$, $2 \times 10^{-1}$, and $2 \times 10^{-3}$ parasite genome equivalents (GE) per reaction to cover a wide range of target abundance. The LAMP reaction products were analyzed by 2% agarose gel electrophoresis using SYBR Gold staining and tested downstream using CRISPR-Cas12a assays with fluorescence readout (see below).

To facilitate master mix preparation for sample processing, LAMP reactions for the preamplification of *Leishmania* DNA targets were performed using the WarmStart LAMP Kit (DNA & RNA) (NEB; Cat no. E1700). Each reaction contained 12.5 µL of WarmStart LAMP 2X Master Mix, 8 µL of molecular biology grade water, and 2.5 µL of 10X LAMP primer mix for a single target (kDNA or 18S rDNA). Like for the LAMP protocol using the separate Bst enzyme, the six target-specific primers were added at a final concentration of 1.6 µM for forward inner and backward inner primers (FIP/BIP), 0.4 µM for loop forward and loop backward primers (LF/LB), and 0.2 µM for forward external and backward external primers (F3/B3). After that, we carefully added 20 µL of immersion oil over the LAMP mix to create two layers to prevent cross-contamination. Then, 2 µL of DNA template (0.26 – 40 ng, for analytical testing) was added to the first (bottom) layer, corresponding to the LAMP mixture, bringing the final reaction volume to 25 µL. The LAMP reaction was incubated on a heating block at 65°C for 30 min (kDNA) or 50 min (18S rDNA), and then stopped by incubation at 82°C for 3 min.

A positive control (*L. braziliensis* M2904 gDNA, $10^2 – 10^4$ GE as indicated in the figure legends) and a negative amplification control (LAMP reaction without DNA template, i.e., No-Template Control (NTC)) were included in all experiments.

## CRISPR-Cas12a assay with fluorescence readout

LbCas12a-based detection reactions were performed as described previously [45,65,66], with the only modification being that a different reporter probe was used, which allows fluorescence signal visualization on a LED transilluminator. We prepared the ribonucleoprotein (RNP) complex 10X by mixing recombinant LbCas12a (produced in-house [67]) with the target-specific folded crRNA (incubated at 65°C for 10 min followed by room temperature (RT) incubation for 10 min) in a 1:1.5 molar ratio and 2 µM single-stranded DNA (ssDNA) reporter probe (/56-FAM/TTATT/3IABkFQ/; [24]) in reaction buffer 1X (NEBuffer 2.1 recipe, prepared in-house, without adding $MgCl_2$). The labeled fluorophore/quencher probe was purchased from Macrogen Inc. (Seoul, South Korea). A volume of 10 µL of the 10X RNP complex (pre-incubated for 10 min at RT) was loaded into each well of a flat-bottom, black 96-well microplate (Thermo Fisher Scientific, Waltham, MA, United States; Cat. no. 237107). Subsequently, 90 µL of the diluted LAMP product –prepared by diluting 2 µL of the LAMP product in 106 µL of 1X reaction buffer containing 16.7 mM $MgCl_2$– was added. The 100 µL LbCas12a *trans*-cleavage reaction, with a final $MgCl_2$ concentration of 15 mM, was incubated and analyzed in a fluorescence plate reader (Synergy H1 hybrid multi-mode reader; Agilent BioTek, Winooski, VT, United States) for 121 min at 25°C. Fluorescence measurements were recorded every 1 min (λex: 490 nm; λem: 525 nm) with a fluorescence gain setting of 120. For the experiments shown in S3 Fig, measurements used the Varioskan LUX multimode microplate reader (Thermo Fisher Scientific).

## Optimization of the CRISPR-Cas12a-based detection assay coupled with lateral flow readout

The lateral flow assay (LFA) readout was performed using the HybriDetect - Universal lateral flow assay kit (Milenia Biotec GmbH, Giessen, Germany; Cat. no. MGHD 1) according to the manufacturer's instructions. In this format, uncleaved reporter molecules are captured at the first detection line (control line) of the LFA strip, whereas collateral Cas12a cleavage activity generates a signal at the second detection line (test line).

Optimization of the LFA readout was conducted as follows. First, to reduce background signal on the T-line in negative controls, we tested the reporter probe at a final concentration (1X) of 500 nM, 250 nM, and 100 nM per LFA. Second, we compared an in-house 1X reaction buffer without $MgCl_2$ (prepared following the NEBuffer 2.1 recipe) to the HybriDetect universal assay buffer. Third, we tested Cas12a reaction incubation at 25°C or 37°C for 30 or 60 min. Fourth, we evaluated the addition of polyethylene glycol (PEG)-8000 at 2% and 4% final concentration to the completed CRISPR-Cas12a reaction, aiming to increase buffer viscosity and thereby slow lateral flow speed.

### Optimized CRISPR-Cas12a-based lateral flow strip assay for clinical samples and reference strain genomic DNA

After completion of the preamplification step, 2 µL of the LAMP product was mixed with 106 µL of 1X reaction buffer with 16.7 mM $MgCl_2$ in a 1.5 mL microcentrifuge tube. Then, we took 90 µL of the diluted LAMP product in a new microcentrifuge tube and added 10 µL of the 10X pre-incubated RNP complex (100 nM Cas12a: 150 nM crRNA: 1000 nM reporter probe). The lateral flow cleavage reporter probe (5'/Bio/TTATTATT/6-FAM/3'; with terminal biotin and FAM labels in the opposite order relative to [26]) was purchased as an HPLC-purified oligo from biomers.net GmbH (Ulm, Germany). The 100 µL LbCas12a *trans*-cleavage reaction, with a final $MgCl_2$ concentration of 15 mM, was incubated at 37°C for 60 min in the dark. Afterward, 5 µL of PEG-8000 (2% final concentration) was added to the reaction tube and homogenized by pipetting up and down. A lateral flow strip was placed into the reaction tube and incubated for 5 min at RT. Lateral flow strips were interpreted visually. LFA strip images shown here were taken with an iPhone 13 Pro camera.

### Analytical sensitivity testing

Serial dilutions of *L. braziliensis* M2904 gDNA (extracted from a promastigote culture) ranging from $2 \times 10^4$ to $2 \times 10^{-3}$ GE per reaction (corresponding to the standard curve of the kDNA qPCR assay [60]) were used as input for LAMP amplification, followed by LbCas12a-based detection with fluorescence and lateral flow readouts. The analytical sensitivity for each target gene was determined based on 3 independent experiments.

### Analytical specificity testing

The analytical specificity of LAMP-CRISPR assays targeting *Leishmania* kDNA (*Viannia* subgenus) or 18S rDNA was tested using gDNA from representative laboratory reference strains of New World and Old World *Leishmania* species, as well as other microbial pathogens (Table 1). A negative control consisting of human gDNA extracted from peripheral blood mononuclear cells (PBMC) of a healthy donor (40 ng of input DNA) was also included. Target genes were amplified by LAMP (using 20–40 ng of input DNA) and detected by LbCas12a assays using fluorescence and lateral flow readouts. Two independent experiments were performed.

### Evaluation of LAMP-CRISPR assay performance on clinical samples

Patient DNA samples extracted from skin lesion specimens were tested blindly by the LAMP-CRISPR assays in groups of ten, plus positive and negative controls. For the LAMP preamplification step, 5 µL of diluted or undiluted DNA (6–285 ng; see S1 Data) was used to capture data from samples with low *Leishmania* parasite load levels. The LAMP products were subsequently detected by LbCas12a assays as described above.

### Data processing and analysis

The data collected from LAMP-CRISPR assays were analyzed as described previously [45]. Briefly, the raw fluorescence data from each well of the LbCas12a assay plate were exported to Microsoft Excel. Based on the inspection of fluorescence time-course data from samples and controls that were run in parallel, the time point of fluorescence accumulation for data analysis was defined (at 25-min time point for both kDNA and 18S rDNA). The

raw fluorescence data were normalized by dividing target reaction fluorescence accumulated at the defined time point of the test sample to that of the NTC reaction included in parallel in the same Cas12a assay plate (further called 'fluorescence ratio'). In the analytical validation, a result was considered 'detected' if a target reaction produced a fluorescence ratio ≥ 2 for the test sample above background (NTC). To set a threshold cutoff value for positive signal (i.e., detection of *Leishmania*) in clinical samples, we required that the fluorescence ratio be larger than the mean plus three standard deviations of negative clinical samples. To this end, a separate panel of confirmed negative samples was used: n = 19 for the kDNA assay and n = 10 for the 18S assay (S1 Data). A larger number of negative samples was used for the kDNA assay to capture inter-assay variation in raw fluorescence signals from the Cas12a reaction observed across different testing periods (S1 Data). Graphs, numerical data analyses, and statistical analyses were performed using GraphPad Prism version 10.2 (GraphPad Software, San Diego, CA, United States).

For the diagnostic performance evaluation of the LAMP-CRISPR assays with fluorescence readout, the sensitivity and specificity and their 95% CIs were calculated with MedCalc software [68] using kDNA real-time PCR as the reference test. To evaluate the accuracy of the LAMP-CRISPR assays to discriminate a positive (detected) from a negative (not detected) *Leishmania* infection status, a receiver operating characteristic (ROC) curve was drawn and the area under the curve (AUC) with its 95% CI was calculated using GraphPad Prism version 10.2. In all statistical tests, a *P* value of < 0.05 was considered statistically significant.

To assess the applicability of the LAMP-CRISPR assays with an LFA readout, 18 clinical samples selected for pilot testing were arbitrarily categorized into three groups based on parasite load: low (<100 parasites per $10^6$ human cells), intermediate (100 – 10,000 parasites per $10^6$ human cells), and high (>10,000 parasites per $10^6$ human cells). LFA strips were interpreted blinded to the fluorescence results.

## Results

### Design and optimization of LAMP-CRISPR assays

We developed two assays for the detection of *Leishmania* spp., targeting the multicopy genetic markers kDNA minicircles and 18S rDNA. These assays combine LAMP with CRISPR-LbCas12a-mediated DNA target recognition and ssDNA reporter cleavage, with results assessed either by fluorescence or lateral flow strip readout (Fig 1). The two readouts are enabled by distinct dual-labeled ssDNA reporters: one labeled with a fluorophore-quencher pair for real-time monitoring of the fluorescence signal on a plate reader, and the other labeled with biotin and FAM for visual detection on paper-based lateral flow strips [26].

We designed LAMP primers flanking reported crRNA target recognition sites [45] within *Leishmania* kDNA minicircles (S1 Fig) and 18S rDNA (S2 Fig). According to *in silico* analyses, the kDNA primer set targets a region most conserved within the *L.* (*Viannia*) subgenus and less conserved within the *L.* (*Leishmania*) subgenus (S1 Fig and S1 File). *In silico* analysis of the selected 18S primer set confirmed the expected conserved nature of the targeted genomic region at the *Leishmania* genus level, whereas sequence variations were more frequent within the *Trypanosoma* genus (S2 Fig and S2 File).

A schematic depicting the mechanism of LAMP amplification, with the resulting LAMP amplicon containing the Cas12a crRNA target sequence is shown in Fig 2. We optimized the LAMP assay reaction conditions using a Bst 2.0 DNA polymerase and tested the incubation time required for optimal amplification of the target DNA. The CRISPR-Cas12a assay was used as the readout. For kDNA, the LAMP reaction was evaluated at 4 incubation times (10, 20, 30, and 40 min) (S3A-S3D Fig). While 10 min were sufficient to amplify the target DNA sequence present at a high quantity ($2 \times 10^2$ GE) (S3A Fig), at least 20 min were required for the amplification of low abundant target DNA ($2 \times 10^{-1}$ GE) (S3B Fig). These results were corroborated by agarose gel electrophoresis (S3F Fig). We chose an incubation time of 30 min as optimal for the kDNA LAMP reaction (S3C and S3E Fig) for further experiments. For 18S rDNA, we tested the LAMP reaction at 5

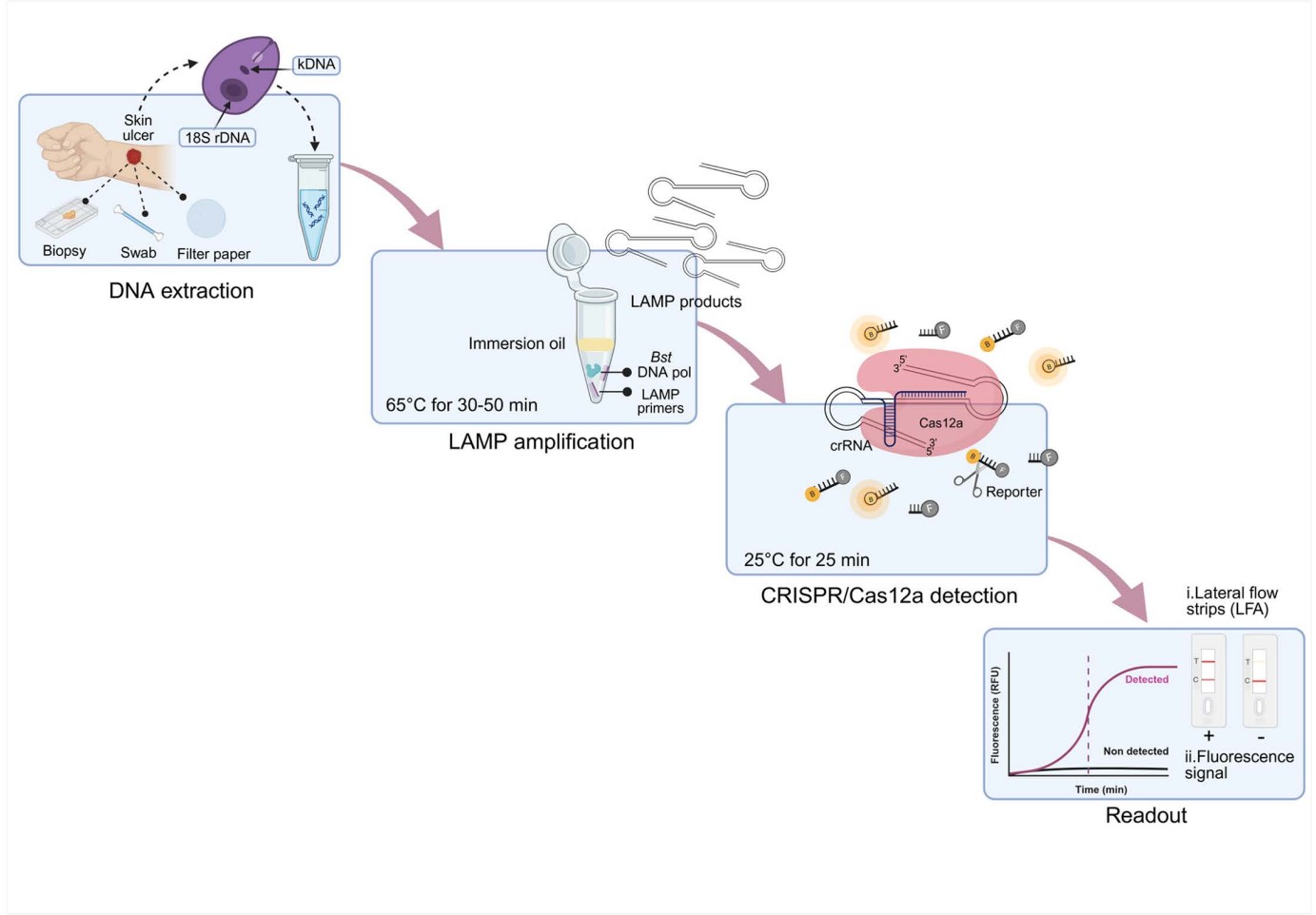

**Fig 1. Schematic representation of the workflow of LAMP-CRISPR assays for detection of *Leishmania* spp. in clinical samples.** Sample preparation is performed from skin lesion specimens taken from patients with suspected CL. DNA samples are subjected to LAMP amplification for 30 min (kDNA) or 50 min (18S rDNA), followed by Cas12a-based reactions with either fluorescence or lateral flow strip readout. Figure created in BioRender. Upc, C. (2026) https://BioRender.com/rpoj3uw, with permission to sublicense under CC-BY 4.0.

incubation times (10, 30, 40, 50, and 60 min) (S3G-S3K Fig). At least 40 min were required to amplify the target DNA (S3I Fig). In both assays, extending the incubation time of the LAMP reaction by 20 min for each target resulted in amplification in the NTC controls (S3D and S3K Fig), indicative of contamination. CRISPR-Cas12a-based detection of LAMP amplicons could effectively discriminate between specific and non-specific amplification. The latter was observed, for instance, at the incubation times of 10 min and 40 min in the NTC reaction by agarose gel electrophoresis (S3M Fig) but did not result in false positives in the Cas12a assay (S3G and S3I Fig). On the basis of these results, we selected an incubation time of 50 min for the 18S LAMP reaction (S3J and S3L Fig) for further experiments.

The CRISPR-Cas12a assay is based on purified recombinant LbCas12a [67]. For the fluorescence-based assay, we used the optimized conditions [45,65,66] and a quenched fluorescent ssDNA reporter [24]. With respect to the LFA readout, we verified the expected signal patterns of the lateral flow strips (S4A Fig) by analyzing one positive control and one negative control, confirming the switch from C-line to T-line upon reporter cleavage in the former (S4B-S4F Fig). The optimized conditions for the LFA readout were as follows: reporter probe at a final concentration of 100 nM in our in-house

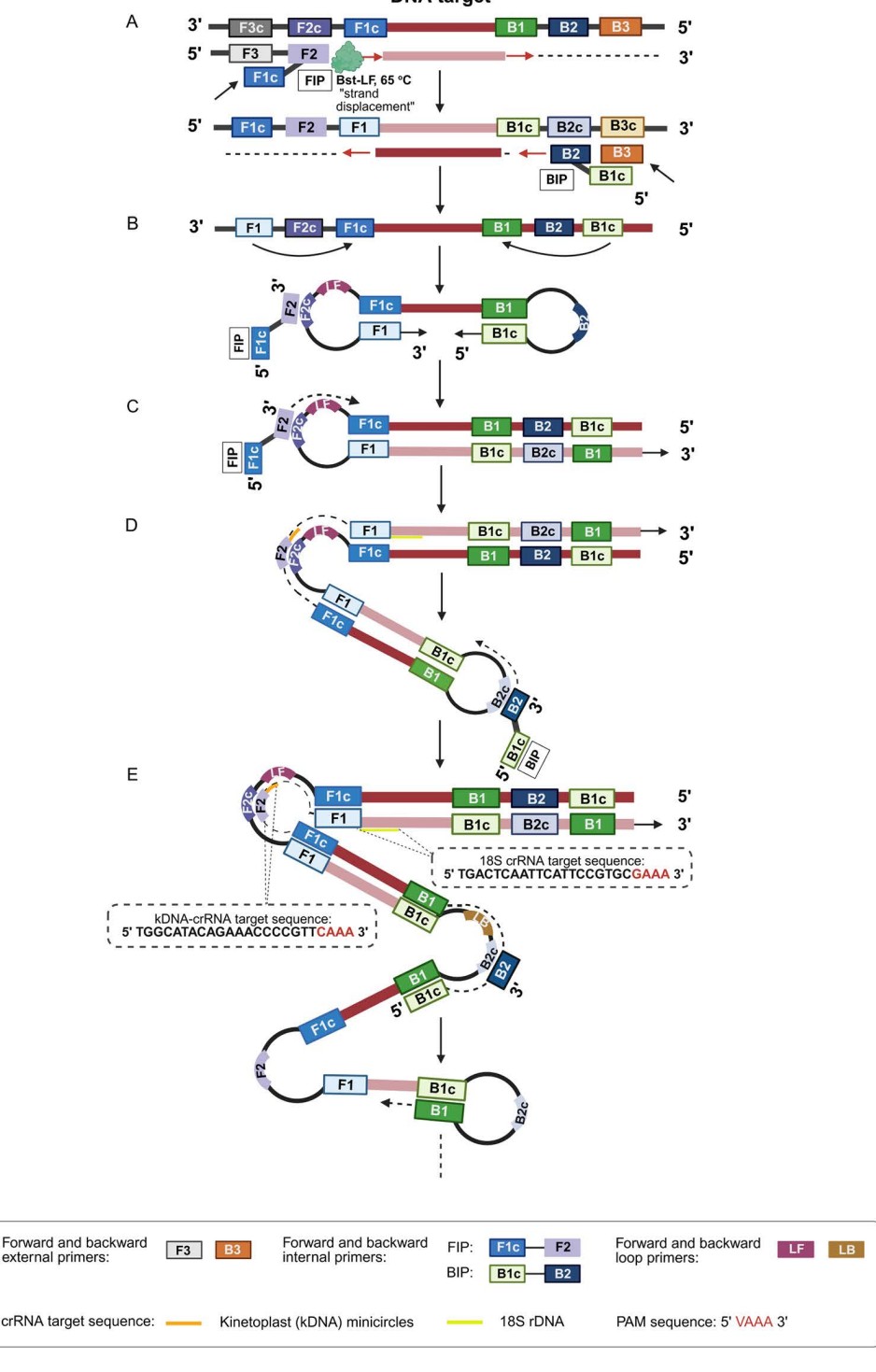

**Fig 2. Schematic representation of the LAMP reaction, highlighting the location of the crRNA target sequence in the target DNA. (A-E)** Steps in the LAMP reaction to amplify the target DNA. The eight annealing locations of the six LAMP primers are shown. The primers were designed to target unique sequences within a conserved region of *Leishmania* kDNA minicircles (S1 Fig) or nuclear 18S rDNA (S2 Fig). The crRNA target sequence and protospacer adjacent motif (PAM) site in the target DNA are shown. For simplicity, the locations of both crRNA target sites are depicted in the same graph. This figure was created de novo by the authors based on the mechanism described in Notomi et al. [14] and Parida et al. [69]. Created in BioRender. Upc, C. (2026) https://BioRender.com/wgofq3n, with permission to sublicense under CC-BY 4.0.

1X Cas12a reaction buffer (which performed similarly to the HybriDetect universal assay buffer); incubation of the Cas12a reaction at 37°C for 60 min; and addition of 2% PEG-8000 to enhance detection (S4B-S4F Fig).

## Analytical validation of LAMP-CRISPR assays

We then assessed the analytical sensitivity of each primer set - crRNA combination using serial dilutions of *L. braziliensis* M2904 gDNA. The kDNA LAMP-CRISPR assay displayed positive *L. braziliensis* detection down to $2 \times 10^{-1}$ GE per reaction, both with a fluorescence readout (Fig 3A) and an LFA readout (Fig 3B). Visualization of the kDNA LAMP products on an agarose gel confirmed the presence of ladder-like bands indicative of a successful amplification (Fig 3C), consistent with the Cas12a-based detection (Fig 3A and 3B). We found a similar analytical sensitivity for the 18S LAMP-CRISPR assay, as detection of at least $2 \times 10^{-1}$ GE per reaction was achieved with both readouts (Fig 3D and 3E). The analysis of the 18S LAMP products by agarose gel electrophoresis (Fig 3F) was consistent with the Cas12a assay results (Fig 3D and 3E). The NTC tested in parallel in each assay showed a background signal in the Cas12a fluorescence-based assay (Fig 3A and 3D, see the two lowest GE input amounts whose fluorescence ratio is 1 given that the denominator is the NTC signal). Likewise, the NTC did not exhibit a band (or the band observed was faint) at the test line in the lateral flow strips (Fig 3B and 3E). This ruled out the possibility of cross-contamination and non-specific amplification.

Specificity testing (S5 Fig) was performed using gDNA samples listed in Table 1. The kDNA LAMP-CRISPR assay consistently detected representative strains of the *L.* (*Viannia*) species present in our sample set (10 of 10; Fig 4A). Additionally, the assay detected 2 out of 6 tested strains belonging to the *L.* (*Leishmania*) subgenus (Fig 4A). The 18S LAMP-CRISPR assay showed pan-*Leishmania* detection ability (Fig 4A). In both assays, no cross-reactivity was observed with related trypanosomatid (*T. cruzi*) strains and other protozoan (*P. falciparum*) and bacterial (*M. tuberculosis*) pathogens tested (Fig 4A). Also, the negative control with human gDNA as template was not detected (Fig 4A). The analytical specificity was also assessed with an LFA readout. Three *L.* (*Viannia*) strains used as positive controls were detected (i.e., showing a strong T-line and a very weak or missing C-line) by both LAMP-CRISPR assays (Fig 4B, strips # 3–5). The negative controls (human gDNA and NTC) had a negative test result (Fig 4B, strips # 1 and 2). There was no cross-reactivity to other microbial pathogens tested (Fig 4B, strips # 6–10, which show a strong C-line and a faint (or absent) T-line, similar to the negative controls).

## Performance evaluation of LAMP-CRISPR assays with clinical samples

Positive patient samples exhibited robust fluorescence curves in the Cas12a assay, indicating the presence of *Leishmania* kDNA (S6A Fig) or 18S rDNA (S6B Fig) molecules. These results were consistent with the respective positive LAMP reactions visualized on agarose gels (S6C and S6D Fig). We noted, however, that agarose gel analysis of LAMP products can give ambiguous results. For instance, ladder-like bands were observed in samples S72 and S09 amplified by the 18S LAMP assay (S6D Fig), but the banding pattern differed from true positive reactions. This was resolved by the Cas12a assay on these amplicons, confirming that S72 and S09 were negative for 18S rDNA (S6B Fig).

We next evaluated the performance of the LAMP-CRISPR assays with a fluorescence readout using gDNA from 90 clinical samples of skin lesions, compared with a reference kDNA qPCR assay. Cutoff values derived from negative samples were 1.46 for the kDNA assay and 1.30 for the 18S assay (S1 Data). The kDNA LAMP-CRISPR assay detected 50 of 55 qPCR-positive samples (90.9%) (Figs 5 and 6A, Table 2), while the 18S LAMP-CRISPR assay detected 40 of 55 (72.7%) (Figs 5 and 6C, Table 2). Both assays achieved 100% specificity (Table 2).

The five qPCR-positive samples not detected by the kDNA LAMP-CRISPR assay (codes 3, 5, 6, 28, 71) harbored low parasite loads (≤20 parasites per $10^6$ human cells), corresponding to $10^{-2}$ to $10^{-3}$ parasite GE per reaction (Fig 6B, S1 Data). Similarly, the 15 qPCR-positive samples not detected by the 18S LAMP-CRISPR assay contained ≤150 parasites per $10^6$ human cells, corresponding to $10^{-1}$ to $10^{-3}$ parasite GE per reaction (Fig 6D, S1 Data). The complete dataset of this study is provided in S1 Data.

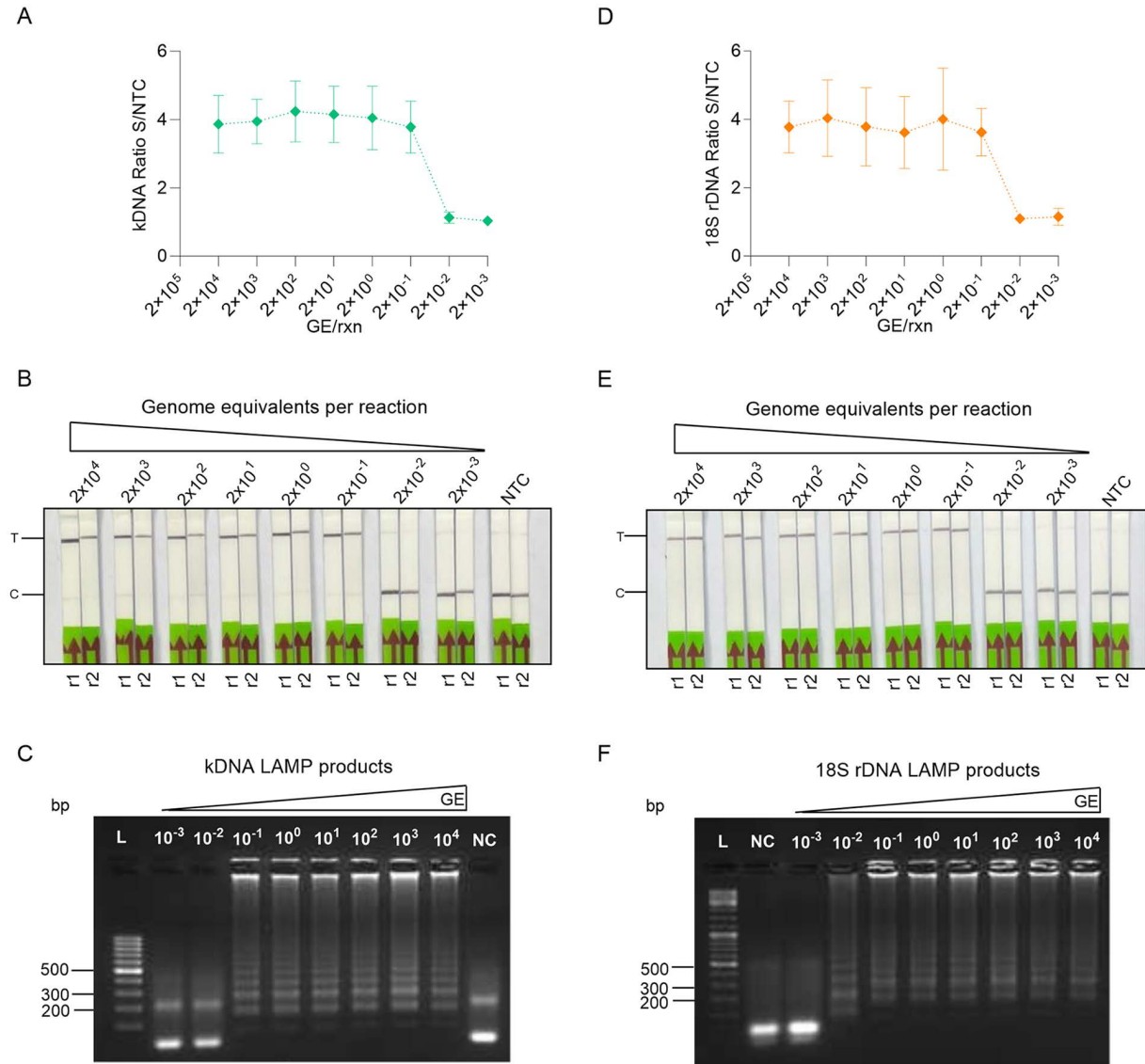

**Fig 3. Analytical sensitivity of kDNA and 18S LAMP-CRISPR assays.** Serial dilutions of *L. braziliensis* M2904 gDNA encompassing $2 \times 10^4$ to $2 \times 10^{-3}$ genome equivalents (GE) per reaction were subjected to LAMP amplification, followed by Cas12a-based reactions for the kDNA (A-C) or 18S rDNA (D-F) target. **(A, D)** Normalized fluorescence signals from Cas12a reactions (i.e., fluorescence signals taken at $t = 25$ min in the test sample relative to the NTC). Data are represented as mean ± SD (n = 3 independent amplification and detection runs). The X-axis has a logarithmic scale. A fluorescence ratio ≥ 2 is considered detected. **(B, E)** Cas12a reactions were performed using a biotin-FAM reporter molecule and visualized on lateral flow strips. The arrow indicates the direction of flow. Two independent amplification and detection runs (r1 and r2) were analyzed. Photos taken by the authors. **(C, F)** LAMP reaction products (5 μL) visualized by 2% agarose gel electrophoresis ran at 100 V for 1 h using SYBR Gold staining. L, GeneRuler 100 bp (C) or 100 bp Plus (F) DNA ladder. NC (negative control): NTC (no-template control).

ROC curve analysis demonstrated good diagnostic performance of the LAMP-CRISPR assays in distinguishing *Leishmania*-positive from -negative clinical samples, with AUC values of 0.973 for the kDNA assay (Fig 7A) and 0.876 for the 18S assay (Fig 7B).

We also assessed the applicability of the LAMP-CRISPR assays with an LFA readout on gDNA extracted from 18 specimens encompassing a wide range of parasite load levels as determined by kDNA qPCR. Lateral flow testing

A

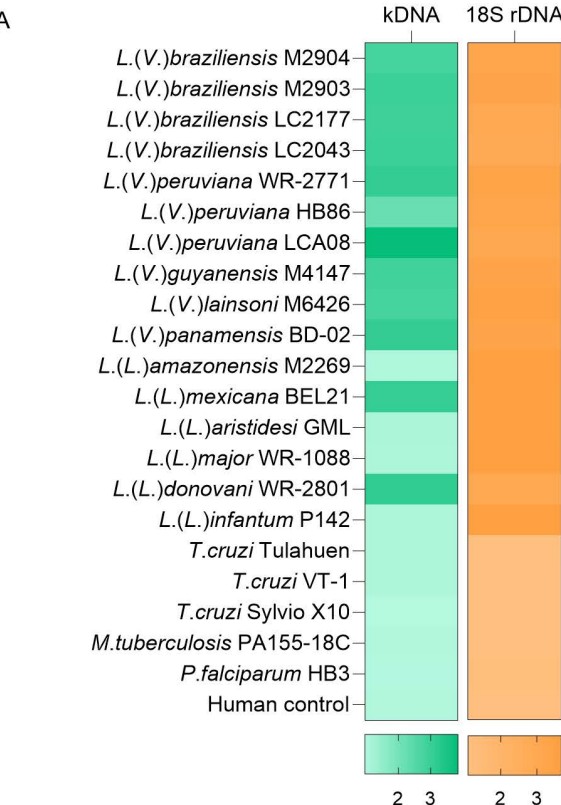

B

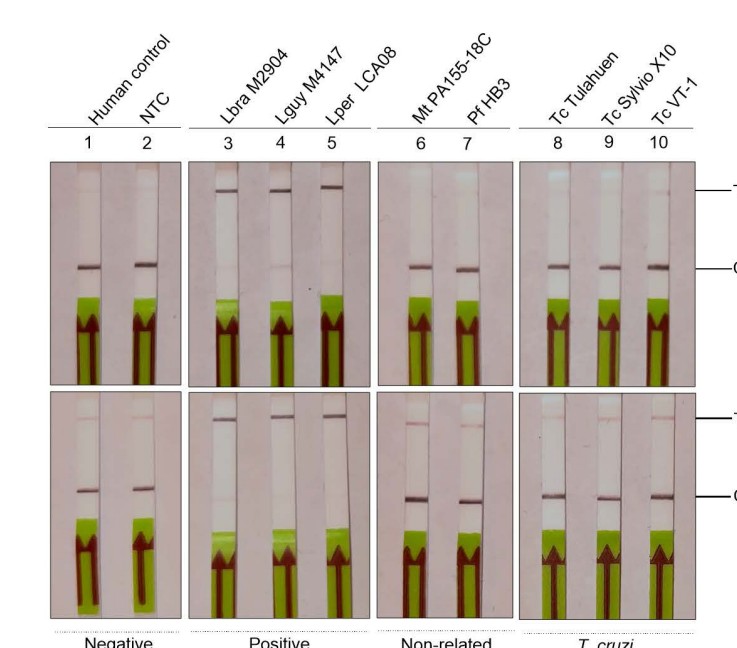

**Fig 4. Analytical specificity of kDNA and 18S LAMP-CRISPR assays.** The specificity of the LAMP-CRISPR assays was evaluated using gDNA samples listed in Table 1. **(A)** Heat maps depicting fluorescence-based detection data. The color scale represents normalized fluorescence signals from Cas12a reactions (n = 2 independent assay repeats). A fluorescence ratio ≥ 2 is considered detected. **(B)** Cas12a reactions with a biotin-FAM reporter

were visualized on lateral flow strips (arrow indicates the direction of flow). Top panel: kDNA assay; bottom panel: 18S assay. Negative controls: HC, human gDNA (strip # 1); NC, no-template control (NTC) (strip # 2). Positive controls: Lbra, *L. braziliensis* M2904 (strip # 3); Lguy, *L. guyanensis* M4147 (strip # 4); Lper, *L. peruviana* LCA08 (strip # 5). Non-related microbes: Mt, *M. tuberculosis* PA155-18C (strip # 6); Pf, *P. falciparum* HB3 (strip # 7). *T. cruzi* (Tc) strains: Tulahuen, Sylvio X10, and VT-1 (strips # 8-10). Photos taken by the authors.

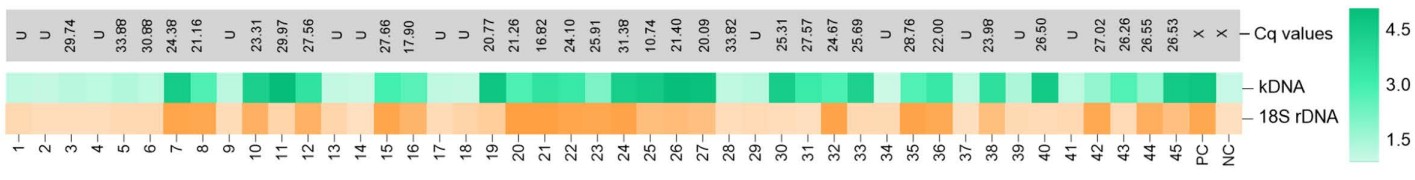

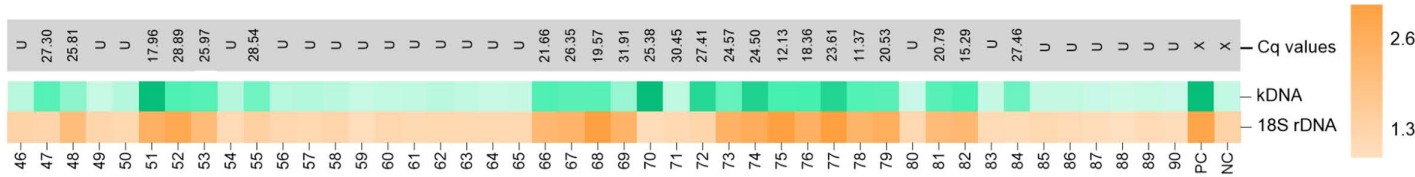

**Fig 5. Performance of kDNA and 18S LAMP-CRISPR assays on clinical samples.** Heat maps showing fluorescence-based detection data from tested clinical samples (n = 90). The color scale represents normalized fluorescence signals from Cas12a reactions. Samples are coded 1-90 ('Code_UPC', S1 Data). PC, positive control (*L. braziliensis* M2904 gDNA, $10^3 – 10^4$ genome equivalents). NC, no-template control. Cutoff values were 1.46 (kDNA) and 1.30 (18S). Cq values from the reference kDNA qPCR assay are shown (U = undetermined, indicating that no Cq values were detected within 35 cycles of amplification; X, not applicable).

results were consistent across two assay repeats per sample (n = 8; S7 Fig), indicating robust assay reproducibility. Of the 18 samples, 16 (88.9%) tested positive by the kDNA LAMP-CRISPR assay, covering the three parasite load categories: low (4 of 6 tested), intermediate (6 of 6), and high (6 of 6) (Fig 8). These LFA results were 100% concordant with the fluorescence-based readout. The 18S LAMP-CRISPR assay with LFA readout detected *Leishmania* DNA in 13 of 18 samples (72.2%), which had parasite load levels defined as: low (2 of 6), intermediate (5 of 6), and high (6 of 6) (Fig 8). One sample (code 12) showed discordant results between LFA and fluorescence-based readouts in the 18S assay, associated with a low parasite load (56.6 parasites per $10^6$ human cells). Two samples from non-leishmaniasis patients (kDNA qPCR-negative; samples 09 and C7) tested negative by both LAMP-CRISPR assays. These results are shown in Figs 8, S7 and S1 Data.

## Discussion

The pressing need for new diagnostic tools has been identified as a priority in the World Health Organization (WHO) 2030 roadmap to support the control and elimination of neglected tropical diseases [70]. Leishmaniasis is among the most neglected, causing a large negative impact on people living in rural endemic areas who are often economically disadvantaged and lack access to an early, correct diagnosis, leading to delayed treatment. PCR-based molecular diagnostics provide the most sensitive and specific tools for *Leishmania* detection and are pivotal for early disease diagnosis, discrimination of *Leishmania* species, detection of asymptomatic infections, and epidemiological surveillance [71]. Beyond their use in centralized laboratories, the progress in the automation and simplification of the workflow from sample preparation to target amplification and detection steps has made molecular diagnostics more suitable and accessible for resource-limited settings in the form of near-PoC tests [72]. Promising formats include the *Leishmania* OligoC-TesT

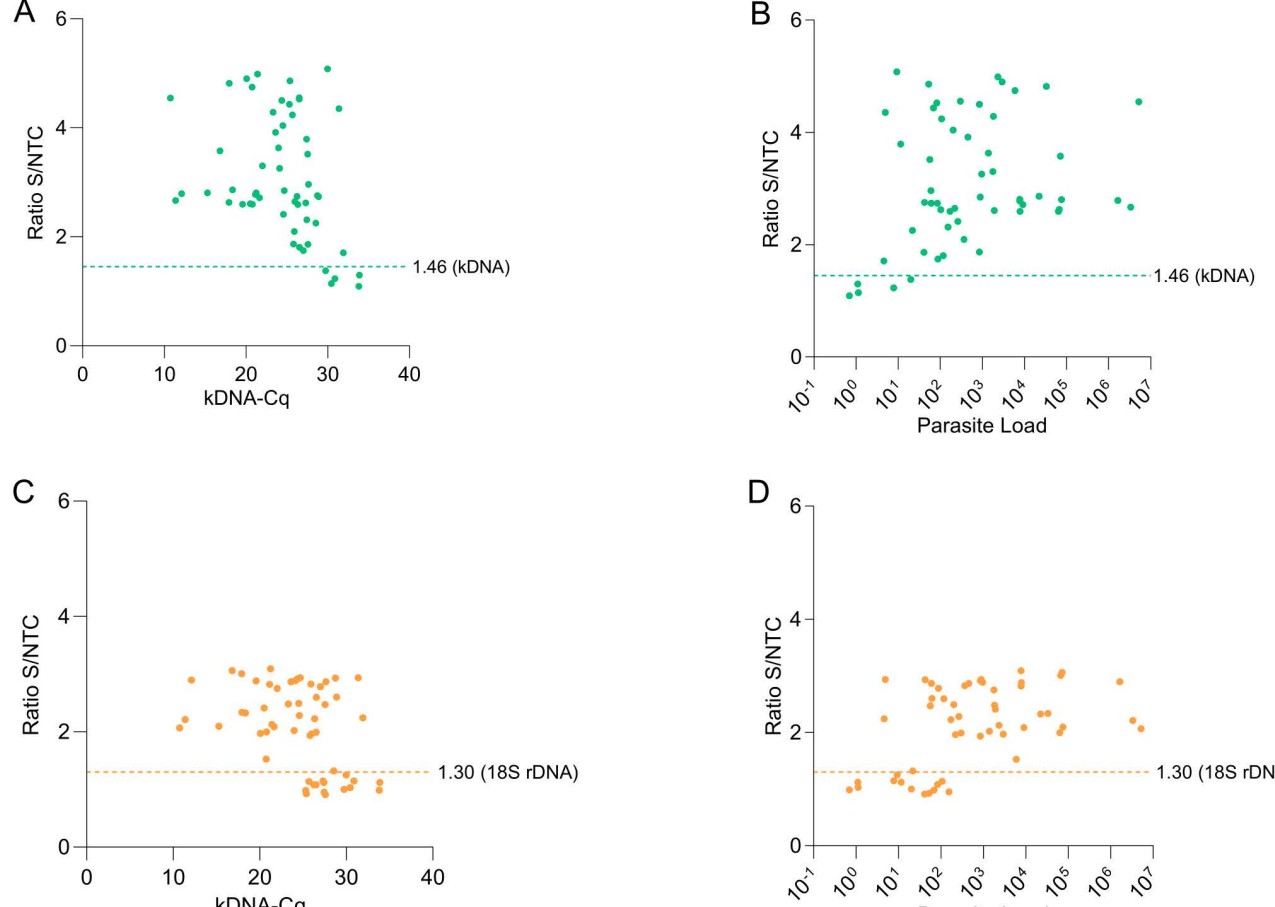

**Fig 6. Performance of kDNA and 18S LAMP-CRISPR assays across parasite load levels in clinical samples. (A, C)** Scatter plots showing normalized fluorescence signals from Cas12a reactions for *Leishmania* kDNA (A) and 18S rDNA (C) in clinical samples versus Cq values from kDNA qPCR (data in Fig 5 and S1 Data). **(B, D)** Scatter plots showing normalized fluorescence signals from Cas12a reactions for kDNA (B) and 18S rDNA (D) versus the estimated parasite load in quantifiable samples (n = 55). The parasite load is expressed as the number of *Leishmania* parasites per $10^6$ human cells. Horizontal dotted lines indicate LAMP-CRISPR assay cutoff values: 1.46 (kDNA) and 1.30 (18S).

**Table 2. Diagnostic performance of LAMP-CRISPR assays with fluorescence readout compared to kDNA qPCR for *Leishmania* DNA detection.**

| | kDNA qPCR | | % (95% CI) | |
|---|---|---|---|---|
| **Test** | **Positive (n = 55)** | **Negative (n = 35)** | **Sensitivity** | **Specificity** |
| kDNA LAMP-CRISPR | | | 90.91 (80.05-96.98) | 100.00 (90.00-100.00) |
| Positive | 50 | 0 | | |
| Negative | 5 | 35 | | |
| 18S LAMP-CRISPR | | | 72.73 (59.04-83.86) | 100.00 (90.00-100.00) |
| Positive | 40 | 0 | | |
| Negative | 15 | 35 | | |

Results of the statistical analysis of diagnostic test performance conducted with the MedCalc software version 23.0.1. (https://www.medcalc.org/calc/diagnostic_test.php)

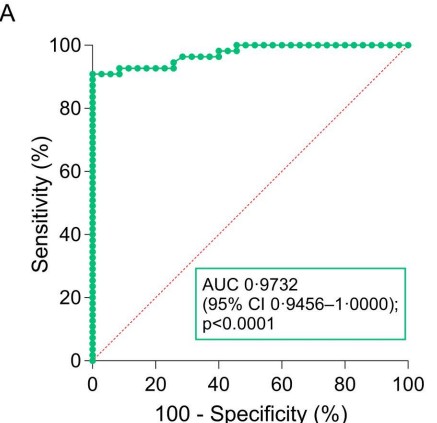
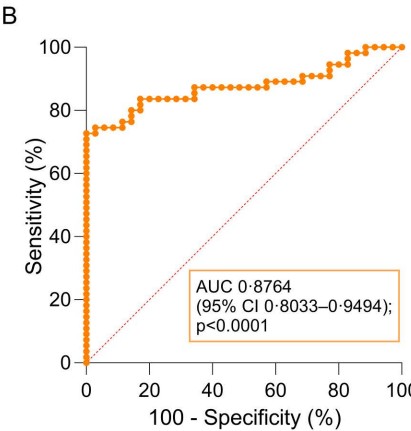

**Fig 7. ROC curve analysis of kDNA and 18S LAMP-CRISPR assays in clinical samples.** ROC curves were generated from fluorescence ratio values of the kDNA **(A)** and 18S **(B)** LAMP-CRISPR assays in 90 clinical samples, using kDNA qPCR as the reference test. The area under the curve (AUC) with 95% CIs (calculated using the Wilson/Brown method) is shown. The diagonal dotted line corresponds to the null hypothesis with AUC of 0.5, in which the accuracy of a diagnostic test is no different from random chance.

(a PCR-oligochromatographic test) [73,74] and a PCR-based method applied directly on EDTA blood for VL diagnosis, using a portable miniPCR device with lateral flow detection [75]. Recent advances also involve isothermal NAATs, such as LAMP and RPA, combined with lateral flow or other portable devices [76–80]. Among the most promising approaches is the integration of isothermal NAATs with CRISPR technology, which has fostered the development of a new generation of molecular diagnostics with the potential to advance toward near-PoC applications in low-resource settings [81].

In this study, we aimed to develop LAMP-coupled CRISPR-Cas12a assays for the molecular detection of *Leishmania* spp. We designed new LAMP primer sets that flank reported crRNA target sites [45] within two commonly used multicopy regions of the *Leishmania* genome: the 18S rDNA and kDNA minicircles. We harnessed the CRISPR-Cas12a system as the detection method to avoid possible false positives caused by non-specific amplification in the LAMP reaction, aiming to enhance assay specificity. The assay can be completed within 1.5 h, with results detectable either through a fluorescence-based readout (compatible with basic molecular biology equipment such as a blue-light transilluminator) or visually on portable lateral flow strips. We report the analytical validation and initial performance evaluation of these assays in clinical samples under controlled conditions in a research laboratory setting. Building on our previous proof-of-concept PCR-CRISPR assays [45], the work presented here lays the groundwork for future efforts aimed at making CRISPR-based diagnostics accessible in a portable testing kit format for the diagnosis of leishmaniasis in rural endemic areas.

Here, in keeping with our *in silico* primer and crRNA design within conserved target genomic regions, the analytical validation demonstrated the ability of the 18S LAMP-CRISPR assay to achieve pan-*Leishmania* detection, whereas the kDNA LAMP-CRISPR assay reliably detected species belonging to the *L.* (*Viannia*) subgenus. The kDNA assay also detected certain strains from the analyzed species of the *L.* (*Leishmania*) subgenus, specifically *L. mexicana* from Central America and *L. donovani* from the Old World. These results reflect both the similarity and variations in mitochondrial kDNA minicircle sequences between the two subgenera (see S1 Fig). Furthermore, our assays showed no cross-reactivity against other pathogens tested, which co-circulate in the same geographic regions and can thus coinfect the same human host, such as *T. cruzi* (a closely related trypanosomatid), *P. falciparum*, and *M. tuberculosis*, or cause cutaneous lesions similar to CL (*M. tuberculosis*) [82]. Our LAMP-CRISPR assays achieved high analytical sensitivity on serially diluted gDNA from the *L. braziliensis* M2904 strain, with a detection level equivalent to 0.2 parasites per reaction. This detection sensitivity (below one parasite GE) compares well with that of the reference kDNA qPCR assay ($5 \times 10^{-3}$ GE per reaction; [60]), which is clinically relevant.

 **Neglected Tropical Diseases**

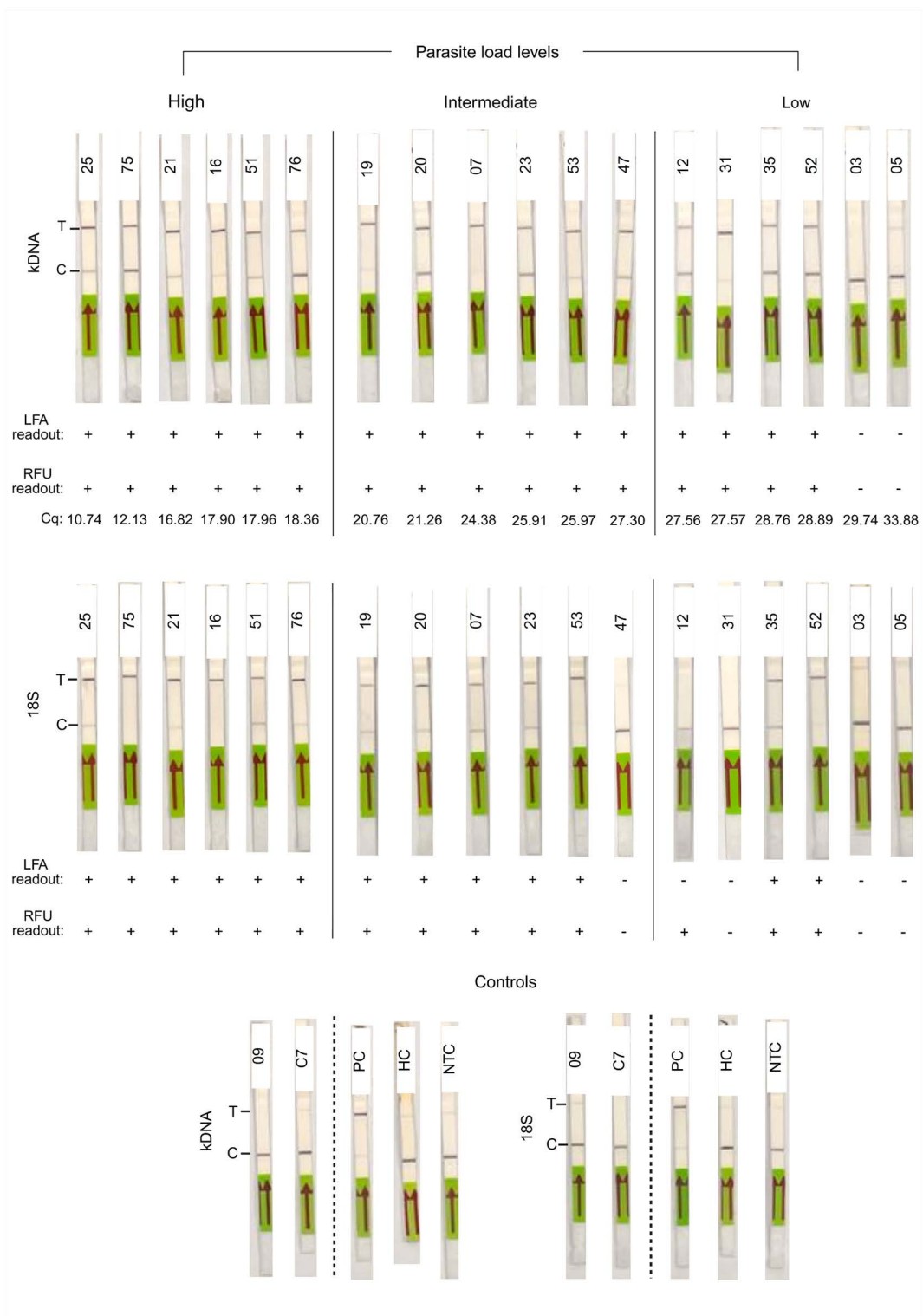

**Fig 8. Pilot testing of kDNA and 18S LAMP-CRISPR assays on clinical samples using a lateral flow readout.** A subset of clinical samples with varying parasite load levels was tested with LAMP-CRISPR assays using an LFA readout. Visualization on strips was achieved through the cleavage of the biotin-FAM reporter. C, control line; T, test line. The fluorescence (RFU) readout (+, detected; -, not detected) as well as Cq values of the kDNA qPCR are shown below each strip. Controls included DNA samples from two non-leishmaniasis patients (qPCR-negative; samples 09 and C7), a positive control (PC, $2 \times 10^2$ GE from *L. braziliensis* M2904 gDNA), a human negative control (HC, 20 ng of PBMC gDNA), and a no-template control (NTC). Photos taken by the authors.

To explore the performance of our LAMP-CRISPR assays to detect *Leishmania* DNA in clinical samples, we evaluated a blinded panel of human skin lesion samples (n = 90) from patients enrolled in the Cusco region of Peru, where CL is caused by *L.* (*Viannia*) species. Both assays with a fluorescence readout attained good diagnostic performance, with sensitivity rates of 90.9% for kDNA and 72.7% for 18S rDNA, and a specificity of 100%, compared to the reference kDNA qPCR assay. Discrepancies between the two methods corresponded to samples with low parasite loads (S1 Data). Overall, LAMP-CRISPR assay results for the subgroup of samples tested with both fluorescence-based and lateral flow strip readouts were concordant. The 18S assay showed a somewhat reduced sensitivity when applied to clinical samples. This is likely related to differences in target copy number: *Leishmania* parasites present an estimated 10–170 copies of the nuclear 18S rDNA region per genome [83,84], while they harbor about 10,000 copies of extrachromosomal kDNA minicircles per cell [85].

Several studies have developed LAMP assays for detecting *Leishmania* infections in diagnosing human VL, CL, and post-kala-azar dermal leishmaniasis (PKDL), as well as in domestic animal reservoirs (dogs) and sand flies. They target multicopy genomic regions, such as kDNA minicircles [86–88], the 18S rRNA gene [89–91], the ITS1 region in the rDNA array [92], cysteine proteinase B genes (*cpb*) [93], and the heat shock protein 70 gene (*HSP70*) [94]. These assays proved to be highly sensitive for the detection of VL- and CL-causing *Leishmania* species (at the genus, species complex, or species level, depending on the target used). To date, few studies have evaluated the diagnostic performance of LAMP assays for New World CL. Nzelu et al. developed a colorimetric LAMP assay using malachite green that targets the *Leishmania* 18S rRNA gene [90]. This assay could reliably amplify *Leishmania* DNA from patients' skin lesion material spotted onto FTA cards [91]. In an evaluation of 122 clinical samples from CL patients in Peru, the FTA-LAMP detected 71/122 (58.2%) samples, performing comparably to nested PCR [91]. León et al. [95] assessed the same LAMP assay [90] on direct smears (n = 50) from suspected CL patients in Colombia, reporting a sensitivity of 100% (95% CI: 98.7-100%) and a specificity of 90.9% (95% CI: 69.4-100%) relative to qPCR. More recently, in Brazil, Soares et al. [94] developed a LAMP assay targeting the *Leishmania HSP70* gene and evaluated its diagnostic accuracy in 100 skin biopsy samples (60 CL cases and 40 non-cases) against qPCR. The LAMP-Leish/HSP70 assay achieved a sensitivity of 86.7% (95% CI: 75.4-94.1%), a specificity of 100% (95% CI: 91.2-100%), and an overall accuracy of 92% (95% CI: 84.8-96.5%). Taken together, these studies underscore the potential of LAMP as a diagnostic and surveillance tool in endemic regions.

With genus-specific targets, because of their high sequence conservation among the Trypanosomatidae, some nonspecific results were observed. Adams et al. [89] reported a LAMP assay based on 18S rDNA that was unable to discriminate between infections with *Leishmania*, *T. brucei*, and *T. cruzi*. Since the diseases caused by these related pathogens differ in clinical presentation, differential diagnosis can be made on clinical symptoms and confirmed using species-specific LAMP assays [89]. In this study, when we tested our 18S LAMP primer set - crRNA combination, no cross-reactivity was observed with DNA from three *T. cruzi* strains belonging to distinct genetic lineages (discrete typing units, DTUs): TcI (Sylvio X10 strain), TcIV (VT-1 strain), and TcVI (Tulahuen LacZ clone C4 strain). These results demonstrate the high analytical specificity of the LAMP-CRISPR assay for *Leishmania* spp. We also noted that integrating LAMP amplification with downstream CRISPR-based detection improved discrimination between positive and negative amplification results, as shown in previous studies [37,56]. This outcome reflects the two layers of specificity intrinsic to LAMP-coupled CRISPR assays: first, the use of 4–6 LAMP primers that selectively target 6–8 distinct regions in the target DNA, and second, the crRNA-guided Cas12a-mediated target sequence recognition [23]. An optimization to improve LAMP assay specificity involves incorporating a probe-specific detection readout [18], as demonstrated by Ruang-Areerate et al. [96] in detecting asymptomatic *Leishmania* infections in HIV-infected patients.

The LAMP technology is commercially available as the Loopamp *Leishmania* Detection Kit (Eiken Chemical Co., Ltd., Japan), which does not require cold-chain storage. This kit targets kDNA minicircles and 18S rDNA in the same reaction to enable pan-*Leishmania* detection based on the LAMP assay developed by Adams et al. [97]. The diagnostic performance evaluation of this kit in different endemic areas around the world has demonstrated its high sensitivity and specificity for

the diagnosis of VL [22,77,97–99], CL [22,97,100,101], and PKDL [102], underscoring its potential for deployment in field settings. So far, two studies have evaluated the efficacy of the Loopamp assay for CL diagnosis in the New World. Adams et al. [97] conducted a prospective cohort study on 105 clinically suspected CL patients in Colombia. Using lesion swab samples, the Loopamp assay achieved a sensitivity of 95% (95% CI: 87.2-98.5%) and a specificity of 86% (95% CI: 67.3-95.9%) compared with a composite reference standard of microscopy and/or culture. In Suriname, Schallig et al. [101] assessed 93 suspected CL cases. The assay had a sensitivity of 84.8% (95% CI: 75.0-91.9%) compared to microscopy and 91.4% (95% CI: 83.0-96.5%) compared to PCR. Specificity was moderate against microscopy (42.9%, 95% CI: 17.7-71.1%) but substantially higher when compared with PCR (91.7%, 95% CI: 61.5-99.8%).

Undoubtedly, the DNA extraction method can impact LAMP assay performance when testing clinical samples. As shown in the study by Ghosh et al. [102], a lower efficiency of LAMP to diagnose PKDL was observed when the LAMP reaction was performed using crude DNA obtained by a simple in-house boil and spin protocol compared to DNA extracted by a silica column-based (Qiagen) kit (67.2% vs. 89.7% sensitivity, respectively). Further, a recent study found a compromised sensitivity (48.4%) of the Loopamp kit for CL diagnosis in Ethiopia, possibly due to a primer mismatch with the *L. aethiopica* 18S rDNA target [103]. This finding points out the need to extensively validate existing LAMP assays in diverse clinical contexts and epidemiological scenarios in different geographical areas to confirm their effectiveness in detecting local parasite variants that are circulating. The same applies to our new LAMP-CRISPR assays.

Combining highly sensitive isothermal amplification with the highly specific CRISPR-Cas12a system has been applied to detect pathogenic trypanosomatids. Wiggins et al. [46] developed proof-of-concept RPA-CRISPR assays that target kDNA maxicircles for *Leishmania* spp. detection, and later adapted them into a multichambered, electricity-free device for diagnostic applications in low-resource settings [104]. Yang et al. [47] developed a one-tube RPA-CRISPR assay that targets the *Leishmania Kmp11* gene. The assay can be completed within 50 min, with the fluorescent signal of reaction tubes observed visually under blue light illumination. The assay yielded a sensitivity of 97.9% and a specificity of 100% on simulated samples. Ortiz-Rodríguez et al. [50] developed a PCR/RPA-CRISPR platform for *T. cruzi* detection. Of three nuclear target genes evaluated, cytochrome B (*Cytb*), the 18S rRNA gene, and histone H2A, only *Cytb* resulted in a workable assay since no mutations were found in the chosen target site during testing of *T. cruzi* DNA extracted from the intestine of triatomine vectors. Furthermore, the authors designed a portable prototype fluorescence reader (TropD-Detector) suitable for low-resource settings. Gutierrez Guarnizo et al. [56] designed a LAMP-CRISPR assay targeting the nuclear *HSP70* gene for *T. cruzi* detection in acute Chagas disease. When evaluated on samples from 100 infants born to Chagas-positive mothers in Bolivia, the assay achieved a sensitivity of 77.27% and a specificity of 100% compared to a TaqMan-based qPCR assay targeting satellite DNA.

Our study shows that LAMP-coupled CRISPR-Cas12a assays are an effective alternative for detecting *Leishmania* infections in various non-invasive and invasive specimens (S1 Data) obtained from human skin lesions. While previous kDNA LAMP primers targeted either the *L. donovani* complex [97] or *L. amazonensis* [88], our kDNA LAMP-CRISPR assay (specifically the crRNA design) exploited a kDNA minicircle region conserved in New World *L.* (*Viannia*) species ( [45] and S1 Fig). In addition, our pan-*Leishmania* 18S LAMP-CRISPR assay is expected to detect species from both the New and Old World (S2 Fig). The performance of these CRISPR-based assays, as demonstrated herein, supports future studies to validate them on a broader scale across diverse geographical regions. Currently, our proof-of-concept workflow requires controlled laboratory conditions. Future work should consider adapting these assays to meet the ASSURED/REASSURED criteria for a diagnostic test suitable for adoption across different healthcare levels, which require assays to be affordable, sensitive, specific, user-friendly, robust, rapid, as well as equipment-free (or operating with minimal equipment), deliverable, with real-time connectivity, and workable on easily collected specimens [105,106]. A major challenge for the implementation of molecular diagnostics in resource-limited settings is sample preparation [107]. Therefore, optimizing rapid DNA extraction protocols, such as boil-spin [102] and SpeedXtract (a magnetic bead-based method) [108], combined with a streamlined CRISPR assay workflow, will be key for advancing toward a near-PoC nucleic acid-based diagnostic assay for CL and other forms of

leishmaniasis. Both LAMP and CRISPR reactions are compatible with freeze-drying, which enhances reagent stability for easy transport and storage at room temperature and may reduce contamination risks [109]. This approach could enable a pre-designed kit format for portable assay setup in resource-limited settings, though its impact on assay performance must be evaluated. Incorporating these refinements alongside user-friendly devices, such as LFA strips or a blue-light transilluminator, would facilitate adoption in minimally-equipped laboratories in endemic regions, and even in a mobile suitcase laboratory. The latter approach has been explored with an RPA assay in field studies conducted in Asia, showing promising performance for VL, PKDL, and CL diagnosis [108,110,111]. Another challenge for implementing these assays in field settings is ensuring continuous training and competency of personnel, particularly given the high turnover among staff. To maintain reliable performance, training programs should not only cover routine test procedures but also emphasize quality control and troubleshooting skills [112]. These training needs can be greatly reduced if the reaction setup and signal readout are integrated into a self-contained, user-friendly device [81]. Another critical factor is the cost of CRISPR-based assay development and validation. The estimated per-sample cost for our laboratory LAMP-CRISPR assays was USD 13.4 with an LFA readout and USD 11.2 with a fluorescence readout, compared to USD 18.3 for kDNA qPCR (see S2 Table). The largest cost component (USD 3.2 per reaction) derives from commercial enzymes in the LAMP master mix kit, which could be reduced through scaling. One alternative is large-scale local production of recombinant enzymes for LAMP and CRISPR reactions; however, this would require additional investment in lot-to-lot analytical validation to ensure consistent reagent performance and to meet regulatory standards for *in vitro* diagnostic applications. The cost of the Milenia HybriDetect LFA kit (USD 2.7 per test) may be brought down through high-volume procurement. In this context, a cost-effective strategy may involve sourcing reagents and consumables in large quantities through supply chains embedded within the healthcare system. Ensuring sustained availability and affordability in low- and middle-income countries will require the integration of CRISPR-based tools and other molecular diagnostics into national health policies.

After further optimization, streamlining, and validation, we envisage that our assays may become valuable tools for providing rapid testing results directly in affected regions, reducing delays associated with testing in centralized laboratories and supporting real-time epidemiological surveillance. Looking ahead, integrating our assays with lab-on-chip systems, such as the paper-based LAMP-CRISPR integrated diagnostic platform (PLACID) [37], could significantly enhance field applicability. PLACID uses freeze-dried reagents and paper microfluidics to perform amplification and detection without fluid transfer, with fluorescence readout via smartphone. This platform showed robust performance for bacterial and viral targets without false positives [37]. These refinements represent important considerations for future assay development. Indeed, our current two-step LAMP-CRISPR open assay format carries an inherent risk of cross-contamination during sample transfer from tube-based (LAMP) to plate-based (CRISPR) reactions, as reported in other studies [109].

In conclusion, we developed novel LAMP-CRISPR assays for the molecular detection of *Leishmania*, targeting two multicopy genomic regions: 18S rDNA for pan-*Leishmania* detection and a kDNA minicircle region conserved among New World *L.* (*Viannia*) species. By integrating LAMP with CRISPR-Cas12a technology, the assays enable accurate detection, with results readout via fluorescence or lateral flow strips, the latter aimed at facilitating a simpler workflow. Analytical validation and initial performance testing on clinical samples demonstrated high sensitivity and specificity, underscoring the assays' potential as effective alternatives for detecting *Leishmania* infections. With further optimization and validation, these proof-of-concept assays hold promise as next-generation molecular tools with great potential to improve the diagnosis and surveillance of leishmaniasis in endemic regions, thereby supporting One Health strategies for disease control.

## Supporting information

**S1 Fig. Multiple sequence alignment of representative kDNA minicircle sequences from members of the *L.* (*Viannia*) and *L.* (*Leishmania*) subgenera, with LAMP primer binding regions and crRNA target site highlighted.** A total of 24 *L.* (*Viannia*) and 35 *L.* (*Leishmania*) sequences were aligned using Clustal Omega (alignments were performed separately for each subgenus) and visualized in Jalview v2.11.4.1. The locations of primer binding regions as well as of

the crRNA target site and PAM sequence in the target DNA are highlighted with colors to indicate sequence conservation within the *L.* (*Viannia*) subgenus and sequence divergence within the *L.* (*Leishmania*) subgenus. Nucleotide variants are represented by spaces in the alignment. The kDNA minicircle sequence from the *L.* (*V.*) *braziliensis* M2904 strain (Gen-Bank accession no. KY698803.1) exhibited 100% identity across all primer and crRNA target regions. The sequences shown are from selected representative strains/isolates of a given *Leishmania* species; additional species and sequences can be examined in the full alignments included in the S1 File. Figure created in BioRender. Upc, C. (2026) https://BioRender.com/0hntnbn, with permission to sublicense under CC-BY 4.0.
(TIF)

**S2 Fig. Multiple sequence alignment of representative 18S rDNA sequences from members of the *Leishmania* and *Trypanosoma* genera, with LAMP primer binding regions and crRNA target site highlighted.** A total of 31 sequences (16 from *Leishmania* spp. and 15 from *Trypanosoma* spp.) were aligned using Clustal Omega and visualized in Jalview v2.11.4.1. Primer binding regions as well as the crRNA target site and PAM sequence in the target DNA are highlighted with colors to indicate sequence conservation within the *Leishmania* genus and sequence variations within the *Trypanosoma* genus. Nucleotide variants are represented by spaces in the alignment. The sequences shown are from selected representative strains/isolates of a given species; additional species and sequences can be examined in the full alignments included in the S2 File. Figure created in BioRender. Upc, C. (2026) https://BioRender.com/54b2q2n, with permission to sublicense under CC-BY 4.0.
(TIF)

**S3 Fig. Optimization of the incubation time of the LAMP reaction.** To determine the minimum necessary incubation time of the LAMP reaction for consistent DNA amplification, tests were performed at different reaction times for *Leishmania* kDNA (10, 20, 30, and 40 min) and 18S rDNA (10, 30, 40, 50, and 60 min) targets. The number and text enclosed in a rectangle indicates the LAMP reaction time. The amplification efficiency was assessed using varying amounts of template DNA (*L. braziliensis* M2904 gDNA), corresponding to approximately $2 \times 10^2$, $2 \times 10^{-1}$, and $2 \times 10^{-3}$ genome equivalents. A negative control (NTC) reaction was tested in parallel. Raw fluorescence signals from Cas12a reactions are shown for kDNA (A-D) and 18S rDNA (G-K) for one representative technical replicate. Normalized data (fluorescence ratio at $t = 25$ min) from two technical replicates are shown in panels E (kDNA) and L (18S rDNA). A fluorescence ratio $\geq 2$ is considered detected. Fluorescence measurements in this figure were taken on the Varioskan LUX plate reader. LAMP reaction products (5 µL each) of kDNA (F) and 18S rDNA (M) were analyzed by 2% agarose gel electrophoresis ran at 100 V for 1 h using SYBR Gold staining. L, 100 bp DNA ladder (ABclonal).
(TIF)

**S4 Fig. Optimization of CRISPR-based detection assays using a lateral flow readout.** (A) Design of the Milenia GenLine HybriDetect test strips and readout interpretation. (Left) In the absence of the specific genetic target, the intact biotin-FAM labeled reporter molecules flow to the control line (C-line). This is interpreted as a negative test result. (Right) Upon recognition of the genetic target, the CRISPR RNP complex cleaves the reporter molecules, which flow to the test line (T-line). The signal intensity of the C-line is weakened. If cleavage of the reporter molecules is partial, both the C-line and T-line may appear with comparable intensity. Both scenarios are interpreted as a positive test result. Figure adapted from [113] and created in BioRender. Upc, C. (2026) https://BioRender.com/bju6fw7, with permission to sublicense under CC-BY 4.0. (B-F) Different parameters influencing the LFA readout performance were evaluated to achieve consistent results: reporter concentration (B), assay buffer (C), incubation temperature (D) and incubation time (E) of the Cas12a assay, and the concentration of PEG-8000 (F). One positive control ($2 \times 10^2$ GE from *L. braziliensis* M2904 gDNA) and a negative control (NTC) were tested. Results shown here correspond to the kDNA LAMP-CRISPR assay. Photos taken by the authors.
(TIF)

**S5 Fig. Workflow from sample preparation to LAMP-CRISPR assays for analytical specificity testing.** Figure created in BioRender. Upc, C. (2026) https://BioRender.com/f9m6gf2, with permission to sublicense under CC-BY 4.0. (TIF)

**S6 Fig. Results of LAMP-CRISPR detection with fluorescence readout in a group of clinical samples.** Raw fluorescence curves generated by Cas12a detection of LAMP amplicons of *Leishmania* kDNA (A) and 18S rDNA (B) in a representative group of clinical samples harboring varying parasite load levels (see S1 Data) and ran in the same assay plate per genomic target. Out of the 8 tested samples shown, 6 showed robust fluorescence curves indicating the presence of *Leishmania* kDNA molecules (panel A; samples coded S16, S23, S31, S52, S72, and S75) whereas the 18S rDNA target was detected in 4 of them (panel B; samples coded S16, S23, S52, and S75). The respective LAMP amplicons (5 µL) of kDNA (C) or 18S rDNA (D) were analyzed by 2% agarose gel electrophoresis (100 V, 60 min) using SYBR Gold staining. Controls included a positive control (PC, $5 \times 10^4$ GE from *L. braziliensis* M2904 gDNA), a human negative control (HC, human PBMC gDNA), and a no-template control (NTC). (TIF)

**S7 Fig. Results of LAMP-CRISPR detection with LFA readout in a group of clinical samples.** Clinical samples chosen for this pilot testing (n = 8) were categorized according to the parasite load levels and tested with kDNA and 18S LAMP-CRISPR assays. Lateral flow testing was performed on two assay repeats, R1 and R2, per sample. Visualization on the lateral flow strips was achieved through the cleavage of the biotin-FAM reporter. Following LFA analysis, the strips were placed into the cassette housing for image acquisition. C, control line; T, test line. Controls included DNA samples from two non-leishmaniasis patients (kDNA qPCR-negative; samples 09 and C7), a positive control (PC, $2 \times 10^2$ GE from *L. braziliensis* M2904 gDNA), a human negative control (HC, 20 ng of human PBMC gDNA), and a no-template control (NTC). Photos taken by the authors. (TIF)

**S1 Table. Primer and crRNA template sequences used in this study.** (PDF)

**S2 Table. Estimated costs of laboratory-based LAMP-CRISPR and qPCR assays.** Per-reaction cost estimates (USD) for LAMP-CRISPR and qPCR assays were calculated based on four categories: assay reagents, sample preparation, disposable materials, and instrument costs. Prices from local distributors included import-related costs applicable in Peru. Instrument costs were estimated using straight-line depreciation (10-year useful life; 10% salvage value), assuming 10,000 reactions per year, following Gutierrez Guarnizo et al. [56]. Recombinant LbCas12a production costs were estimated using the reagent information reported by Mendoza-Rojas et al. [67] and the measured enzyme concentration obtained per liter of *E. coli* culture (2.2 µM). LAMP-CRISPR runs included 10 clinical samples plus controls. qPCR runs included 12 samples analyzed in triplicate, along with a standard curve and controls. Per-reaction costs were converted to per-sample costs by prorating controls and accounting for technical replicates. (XLSX)

**S1 File. *In silico* alignment of *Leishmania* kDNA minicircle sequences from *Viannia* and *Leishmania* subgenera, and location of primer and crRNA binding sites.** This file contains sequence alignments and visualizations used to assess the *in silico* specificity of the designed LAMP primers and Cas12a crRNA within conserved regions of *Leishmania* kDNA minicircles. A total of 1,060 aligned sequences from strains/isolates belonging to the *L.* (*Viannia*) subgenus and 965 aligned sequences from strains/isolates of the *L.* (*Leishmania*) subgenus were included. Primer and crRNA binding regions were manually annotated using Jalview v2.11.4.1 to enhance visualization and facilitate identification of conserved target sites. Additionally, a subset of representative sequences, 24 *L.* (*Viannia*) and 35 *L.* (*Leishmania*), is provided in

separate alignment project files to illustrate kDNA minicircle sequence conservation and divergence across the two subgenera (see also S1 Fig).
(ZIP)

**S2 File. *In silico* alignment of 18S rDNA sequences from *Leishmania* and *Trypanosoma* genera, and location of primer and crRNA binding sites.** This file contains the alignments and visualizations used to evaluate the *in silico* specificity of the designed LAMP primers and Cas12a crRNA targeting the 18S rDNA. It includes 37 aligned sequences of *Leishmania* spp. and 272 of *Trypanosoma* spp. Primer and crRNA binding regions were manually annotated using Jalview v2.11.4.1 to enhance visualization and identification. Additionally, an alignment of a subset of 31 representative sequences (16 *Leishmania*, 15 *Trypanosoma*) is provided for visualization of 18S rDNA sequence conservation and divergence across the two genera (see also S2 Fig).
(ZIP)

**S1 Data. Excel file containing the data on tested clinical samples generated during the current study.** This file contains the LAMP-CRISPR and qPCR data on tested clinical samples, as well as the determination of threshold cutoff values for fluorescence-based CRISPR-Cas12a assay result interpretation.
(XLSX)

## Acknowledgments

We thank Pohl Milón (Universidad Peruana de Ciencias Aplicadas) for inspiring discussions; Yomara Romero (Universidad Peruana de Ciencias Aplicadas) for assistance with sequence alignments and advice on their presentation; Edith Málaga and Manuela Verástegui (Universidad Peruana Cayetano Heredia) for providing genomic DNA from the *Trypanosoma cruzi* strains tested here; and the Office of Research at the Universidad Peruana de Ciencias Aplicadas for the support provided to carry out this research work.

## Author contributions

**Conceptualization:** Vanessa Adaui.

**Formal analysis:** Eva Dueñas, Percy Huaihua, Vanessa Adaui.

**Funding acquisition:** Vanessa Adaui.

**Investigation:** Eva Dueñas, Ingrid Tirado, Percy Huaihua, Ariana Parra del Riego, Luis Cabrera-Sosa, Jose A. Nakamoto, Carlos M. Restrepo.

**Project administration:** Vanessa Adaui.

**Resources:** Luis Cabrera-Sosa, Jose A. Nakamoto, María Cruz, Carlos M. Restrepo, Jorge Arévalo, Vanessa Adaui.

**Supervision:** Jorge Arévalo, Vanessa Adaui.

**Visualization:** Eva Dueñas, Ingrid Tirado.

**Writing – original draft:** Eva Dueñas, Ariana Parra del Riego, Vanessa Adaui.

**Writing – review & editing:** Eva Dueñas, Ingrid Tirado, Percy Huaihua, Ariana Parra del Riego, Luis Cabrera-Sosa, Jose A. Nakamoto, María Cruz, Carlos M. Restrepo, Jorge Arévalo, Vanessa Adaui.

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
