## [Decision Letter · Decision Letter 0]

29 Oct 2025

*Leishmania*
Response to Reviewers
Revised Manuscript with Track Changes
Manuscript

Shaden Kamhawi

co-Editor-in-Chief

Paul Brindley

co-Editor-in-Chief

**Additional Editor Comments :**
**Journal Requirements:**

1) We do not publish any copyright or trademark symbols that usually accompany proprietary names, eg ©,  ®, or TM  (e.g. next to drug or reagent names). Therefore please remove all instances of trademark/copyright symbols throughout the text, including:

- ® on pages: 10, and 13

- TM on pages: 6, 15, and 35.

2) Some material included in your submission may be copyrighted. According to PLOSu2019s copyright policy, authors who use figures or other material (e.g., graphics, clipart, maps) from another author or copyright holder must demonstrate or obtain permission to publish this material under the Creative Commons Attribution 4.0 International (CC BY 4.0) License used by PLOS journals. Please closely review the details of PLOSu2019s copyright requirements here: PLOS Licenses and Copyright. If you need to request permissions from a copyright holder, you may use PLOS's Copyright Content Permission form.

Potential Copyright Issues:

i) Please confirm (a) that you are the photographer of 3B, 4B, 8, and S7, or (b) provide written permission from the photographer to publish the photo(s) under our CC BY 4.0 license.

ii) Figures 1, 2, S1, S2, S4, and S5. We note that the figures are created through BioRender. Please confirm that you hold a Premium account and provide a pdf copy of the CC BY 4.0 Licence as provided by BioRender. For instructions on how to generate a CC BY 4.0 license for your figure, please see the guidelines here: https://help.biorender.com/hc/en-gb/articles/21282341238045-Publishing-in-open-access-resources.

If you are using the free assets from BioRender, we are unable to publish these images as they are licenced under a stricter licence than CC BY 4.0. In this case we ask you to remove the BioRender images and replace them with open source alternatives.

See these open source resources you may use to replace images / clip-art:

- https://bioart.niaid.nih.gov/

- https://bioicons.com/

- https://healthicons.org/

- https://scidraw.io/

- https://reactome.org/icon-lib

- https://www.phylopic.org/images

- https://journals.plos.org/plosbiology/article?id=10.1371/journal.pbio.3002395

iii) Thank you for inducationg that Figures 2, and S4 are adapted from previously published figures. Please include in the figures legends direct links to the sources of the figure and provide links to the terms of use / license information.

Note : If the figure is adapted from a copyrighted source, please provide written permission from the copyright holder to publish this under our CC-BY 4.0 license, or remove the figure / replace the image. Please note we do not recommend using standard request forms available on Publishers' websites, as they grant single use rather than republication under an open access license.

3) Please amend your detailed Financial Disclosure statement. This is published with the article. It must therefore be completed in full sentences and contain the exact wording you wish to be published.

1) State the initials, alongside each funding source, of each author to receive each grant. For example: "This work was supported by the National Institutes of Health (####### to AM; ###### to CJ) and the National Science Foundation (###### to AM).".

**Reviewers' comments:**

**Key Review Criteria Required for Acceptance?**

**Methods**

-Are the objectives of the study clearly articulated with a clear testable hypothesis stated?

-Is the study design appropriate to address the stated objectives?

-Is the population clearly described and appropriate for the hypothesis being tested?

-Is the sample size sufficient to ensure adequate power to address the hypothesis being tested?

-Were correct statistical analysis used to support conclusions?

-Are there concerns about ethical or regulatory requirements being met?

Reviewer #1: -Are the objectives of the study clearly articulated with a clear testable hypothesis stated? Yes

-Is the study design appropriate to address the stated objectives? Yes

-Is the population clearly described and appropriate for the hypothesis being tested? Yes

-Is the sample size sufficient to ensure adequate power to address the hypothesis being tested? Yes

-Were correct statistical analysis used to support conclusions? yes

-Are there concerns about ethical or regulatory requirements being met? yes

Reviewer #2: The study objectives, study design, statistical analysis and conclusion are fine. The sample size for clinical evaluation of the tests is not justified.

Reviewer #3: All methods used are suitable for answering the hypothesis.

**Results**

-Does the analysis presented match the analysis plan?

-Are the results clearly and completely presented?

-Are the figures (Tables, Images) of sufficient quality for clarity?

Reviewer #1: -Does the analysis presented match the analysis plan? yes

-Are the results clearly and completely presented? Yes

-Are the figures (Tables, Images) of sufficient quality for clarity? Yes

Reviewer #2: The analysis and presentation of the results are clear and tables and figures as well.

Reviewer #3: All results presented are in line with the analysis plan and are clearly presented.

**Conclusions**

-Are the conclusions supported by the data presented?

-Are the limitations of analysis clearly described?

-Do the authors discuss how these data can be helpful to advance our understanding of the topic under study?

-Is public health relevance addressed?

Reviewer #1: -Are the conclusions supported by the data presented? yes

-Are the limitations of analysis clearly described? yes

-Do the authors discuss how these data can be helpful to advance our understanding of the topic under study? yes

-Is public health relevance addressed? yes

Reviewer #2: The conclusion is supported by the obtained results. Limitation of the study is that it needs further improvement to comply with WHO ASSURED criteria. They, however, need to indicate approximate cost of the current version of the tests and how they plan to bring it down so that it can be implemented in the LMIC if eventual phase III diagnostic trial confirmed tests' diagnostic accuracy more than 95%.

Reviewer #3: All conclusions are corroborated by the data presented and have an impact on public health. The authors discuss the limitations of the analysis.

**Editorial and Data Presentation Modifications?**

Reviewer #1: NA

Reviewer #2: The current version of the manuscript is too long. Repeats can be avoided. Technical details can be summarized where applicable.

Reviewer #3: The manuscript presents robust results on the use of LAMP combined with CRISPR-Cas12a for the diagnosis of cutaneous leishmaniasis. The two assays evaluated, particularly the one using the kDNA target, showed good analytical and clinical performance. My suggestion is that the article be accepted after minor modifications.

Comments:

1- Include in the Methods section the criterion (qPCR kDNA) used to define cases and non-cases of cutaneous leishmaniasis;

2- Do the authors have information on the diseases of the patients in the non-case group? If so, it would be helpful to include this information in the supplementary table;

3- Were parasitological tests (imprint, scarification, or culture) performed on the patients? If so, it would be important to compare the performance of these techniques with the results obtained with the LAMP-CRISPR-Cas12a assays;

4- Although the authors suggest that the CRISPR-Cas12a system increases the specificity of the reaction, it would be interesting to demonstrate the results of the 18S LAMP and kDNA assays without the use of the CRISPR system. This comparative analysis would demonstrate the real need to use CRISPR to increase specificity;

5- Include in the discussion other articles that evaluated Lamp for cutaneous leishmaniasis, without the use of the CRISPR-Cas12a system.

**Summary and General Comments**

Reviewer #1: The manuscript presents the development and validation of CRISPR–Cas12a-based molecular assays coupled with LAMP amplification for the detection of Leishmania spp. The study is well designed, clearly written, and makes a significant contribution to the field of molecular diagnostics for tegumentary leishmaniasis, particularly in the context of low-resource endemic settings. The work is timely and relevant, and the results are convincingly presented. I have only minor comments and suggestions that may help strengthen the manuscript:

Contextualization of field implementation

The authors may wish to expand slightly on potential challenges or considerations for the implementation of these assays in real-world field settings (e.g., sample preparation, training needs, device portability, or cold chain requirements). This would better highlight the practical applicability of the tool.

Clarification of assay comparison

The comparison with reference qPCR methods is clearly described. It could be helpful to add a brief statement on the Ct range or parasite loads of the discordant samples to further illustrate assay performance in low-parasite contexts.

Discussion on 18S assay performance

The lower sensitivity of the 18S assay compared to kDNA is briefly noted. A short discussion of potential biological or technical reasons for this difference (e.g., target copy number, sequence diversity) would be valuable for readers considering assay selection.

Minor editorial points

Ensure consistency in the use of species names (e.g., Leishmania (Viannia) vs. L. (Viannia)).

A few typographical issues were noted (e.g., “crossreactivity” should be “cross-reactivity”). A final language check is recommended.

Overall, this is an excellent manuscript that presents innovative diagnostic tools with clear public health relevance. With minor clarifications, it will make a strong contribution to the literature on Leishmania diagnostics.

Reviewer #2: The study is encouraging to pave the way to take molecular test in a lateral-flow test. However, the proposed DNA amplification tools are challenging in the field which limits its eventual implementation in the LMIC. The sensitivity is

Reviewer #3: Comments above

PLOS authors have the option to publish the peer review history of their article (what does this mean? ). If published, this will include your full peer review and any attached files.

**Do you want your identity to be public for this peer review?** For information about this choice, including consent withdrawal, please see our Privacy Policy .

Reviewer #1: No

Reviewer #2: No

Reviewer #3: No

**Figure resubmission:**

**Reproducibility:** To enhance the reproducibility of your results, we recommend that authors of applicable studies deposit laboratory protocols in protocols.io, where a protocol can be assigned its own identifier (DOI) such that it can be cited independently in the future. Additionally, PLOS ONE offers an option to publish peer-reviewed clinical study protocols. Read more information on sharing protocols at https://plos.org/protocols?utm_medium=editorial-email&utm_source=authorletters&utm_campaign=protocols

---

## [Decision Letter · Decision Letter 1]

22 Jan 2026

Dear Mrs. Adaui,

We are pleased to inform you that your manuscript 'LAMP-coupled CRISPR-Cas12a assays: a promising new tool for molecular diagnosis of leishmaniasis' has been provisionally accepted for publication in PLOS Neglected Tropical Diseases.

Best regards,

Alain Debrabant

Academic Editor

Sarman Singh

Section Editor

Shaden Kamhawi

co-Editor-in-Chief

Paul Brindley

co-Editor-in-Chief

Your revised manuscript was reviewed by Reviewer 3 and the Editors and found acceptable for publication in PNTD.

Reviewer's Responses to Questions

**Key Review Criteria Required for Acceptance?**

**Methods**

-Are the objectives of the study clearly articulated with a clear testable hypothesis stated?

-Is the study design appropriate to address the stated objectives?

-Is the population clearly described and appropriate for the hypothesis being tested?

-Is the sample size sufficient to ensure adequate power to address the hypothesis being tested?

-Were correct statistical analysis used to support conclusions?

-Are there concerns about ethical or regulatory requirements being met?

Reviewer #3: Yes

**Results**

-Does the analysis presented match the analysis plan?

-Are the results clearly and completely presented?

-Are the figures (Tables, Images) of sufficient quality for clarity?

Reviewer #3: Yes

**Conclusions**

-Are the conclusions supported by the data presented?

-Are the limitations of analysis clearly described?

-Do the authors discuss how these data can be helpful to advance our understanding of the topic under study?

-Is public health relevance addressed?

Reviewer #3: Yes

**Editorial and Data Presentation Modifications?**

Reviewer #3: Accept

**Summary and General Comments**

Reviewer #3: The authors made all the requested changes.

PLOS authors have the option to publish the peer review history of their article (what does this mean? ). If published, this will include your full peer review and any attached files.

**Do you want your identity to be public for this peer review?** For information about this choice, including consent withdrawal, please see our Privacy Policy .

Reviewer #3: **Yes:** Daniel Moreira de Avelar

---

## [Editor Report · Acceptance letter]

Dear Mrs. Adaui,

We are delighted to inform you that your manuscript, "LAMP-coupled CRISPR-Cas12a assays: a promising new tool for molecular diagnosis of leishmaniasis," has been formally accepted for publication in PLOS Neglected Tropical Diseases.

Best regards,

Shaden Kamhawi

co-Editor-in-Chief

Paul Brindley

co-Editor-in-Chief
